# An adaptation of astronomical image processing enables characterization and functional 3D mapping of individual sites of excitation-contraction coupling in rat cardiac muscle

Qinghai Tian, Lars Kaestner, Laura Schröder, Jia Guo, Peter Lipp*

Institute for Molecular Cell Biology, Center for Molecular Signaling (PZMS), Medical Faculty, Saarland University, Homburg, Germany

**Abstract** In beating cardiomyocytes, synchronized localized $Ca^{2+}$ transients from thousands of active excitation-contraction coupling sites (ECC couplons) comprising plasma and sarcoplasmic reticulum membrane calcium channels are important determinants of the heart's performance. Nevertheless, our knowledge about the properties of ECC couplons is limited by the lack of appropriate experimental and analysis strategies. We designed CaCLEAN to untangle the fundamental characteristics of ECC couplons by combining the astronomer's CLEAN algorithm with known properties of calcium diffusion. CaCLEAN empowers the investigation of fundamental properties of ECC couplons in beating cardiomyocytes without pharmacological interventions. Upon examining individual ECC couplons at the nanoscopic level, we reveal their roles in the negative amplitude-frequency relationship and in β-adrenergic stimulation, including decreasing and increasing firing reliability, respectively. CaCLEAN combined with 3D confocal imaging of beating cardiomyocytes provides a functional 3D map of active ECC couplons (on average, 17,000 per myocyte). CaCLEAN will further enlighten the ECC-couplon-remodelling processes that underlie cardiac diseases.

DOI: https://doi.org/10.7554/eLife.30425.001

*For correspondence:
peter.lipp@uks.eu

**Competing interests:** The authors declare that no competing interests exist.

## Introduction

Cardiac myocytes are the functional cellular units of the heart, whose contractile properties largely determine the pumping performance of the heart (*Bers, 2002*). During cellular action potentials, voltage-operated L-type $Ca^{2+}$ channels (LTCCs) in the plasma membrane evoke a brief $Ca^{2+}$ influx that is amplified by ryanodine receptors (RyRs) in the membrane of the sarcoplasmic reticulum (SR). This process is referred to as $Ca^{2+}$-induced $Ca^{2+}$ release (CICR) (*Bers, 2002*). It is generally accepted that LTCCs and RyRs are clustered on their respective membranes and oppose each other strongly to build up the functional excitation-contraction coupling (ECC) unit, the so-called ECC couplon (*Soeller et al., 2007*). For each heartbeat, reliable communication between these partners ensures robust functional coupling of the plasma and SR membranes and enables synchronized activation of ECC couplons throughout the myocyte, a prerequisite for optimal contractility of the cell (*Wier and Balke, 1999*). Many processes contribute to the dynamics of $Ca^{2+}$ cycling in cardiomyocytes, including the opening of LTCCs, the activity of the sarco/endoplasmic reticulum $Ca^{2+}$-ATPase (SERCA) and the functional state of RyRs (*Bers, 2008*).

The adrenergic hormone system is a major regulator of minute-by-minute cardiovascular function (*Dorn, 2010*) and evokes activation of the cAMP/protein kinase A signalling pathway, resulting in

the phosphorylation of key proteins of ECC, including LTCCs, phospholamban, and RyRs. Ultimately, the system results in an increased cellular $Ca^{2+}$ transient and in cellular force production, fostering the mechanical output of the heart and thus providing a positive inotropic response (*Bers, 2002*). In addition, β-adrenergic stimulation in vivo exerts a positive chronotropic effect, highlighting the importance of understanding the force-frequency response as an important physiological mechanism regulating cardiac output (*Anderson et al., 1973*). As part of the negative force-frequency relationship found in small rodents (e.g., rat and mouse), the amplitude of the global $Ca^{2+}$ transient decreases with increasing stimulation frequency (*Sipido et al., 1998*; *Gattoni et al., 2016*). Several different processes might contribute to this behavior, including a decreased SR-$Ca^{2+}$ content, a decreased number of 'participating' ECC couplons and decreased $Ca^{2+}$ release from individual ECC couplon sites, all of which are able to diminish the global, cellular $Ca^{2+}$ transient. Whether and to what degree the reliability of couplons' firing adds to the latter contributor remains elusive.

The current knowledge of RyR cluster behavior has substantially increased following the advent of confocal microscopy, which enables the study of spontaneous, local $Ca^{2+}$ release signals, the $Ca^{2+}$ spark (*Cheng et al., 1993*; *Lipp and Niggli, 1994*; *Cheng and Lederer, 2008*). Although ECC couplon sites represent a subset of $Ca^{2+}$ spark sites and they share similar properties, their mode of occurrence–i.e., evoked vs. spontaneous, synchronized vs. uncoordinated—disallows direct transfer of $Ca^{2+}$ spark properties onto those of the couplon site. Localized $Ca^{2+}$ transients from ECC couplons occur in a synchronized manner through the action potential and in great numbers (more than ten thousand within a few tens of milliseconds) (*Bers, 2002*). The analysis of spontaneous $Ca^{2+}$ sparks will not capture the properties of ECC couplons following electrical orchestration because spontaneous openings of RyRs might be under the control of different sets of environmental parameters, and because they lack the signal transduction component of translating the electrical action potential into $Ca^{2+}$ channel opening and into localized $Ca^{2+}$ influx. These complex processes substantially determine a majority of the properties of ECC couplon sites and thus contribute to shaping the global $Ca^{2+}$ transient.

The number and arrangement of LTCC and RyR clusters can be determined by appropriate immunohistochemistry. However, relating conclusions from such preparations to the physiological distribution of electrically evoked and functional ECC couplons (e.g., by co-localization analysis) is rather difficult and often based on many assumptions. One of the most striking problems is that microscopic approaches to studying co-localization, even with super-resolution techniques, are still too coarse to identify functional coupling between these two partners reliably during normal ECC. In addition, fixation artefacts, such as shrinking of the preparation or alteration of the nanoscopic arrangement of proteins, might affect the aforementioned analysis. Thus, it clearly follows that approaches allowing for a robust in situ analysis (i.e., in the living naive cell during normal ECC) are eagerly anticipated not only to provide a full appreciation of the physiological 3D distribution of active ECC couplons but also to reveal pathophysiological alterations therein.

The spontaneous firing of individual $Ca^{2+}$ release clusters, that is $Ca^{2+}$ sparks, can technically be monitored by laser-scanning microscopy (*Cheng et al., 1993*; *Cheng and Lederer, 2008*). However, the firing of thousands of synchronized but ultra-fast and highly dynamic ECC couplons renders it almost impossible to investigate the behavior of individual ECC couplons during physiological ECC in the absence of pharmacological or experimental interventions such as high $Ca^{2+}$ buffer concentrations (*Cleemann et al., 1998*; *Woo et al., 2002*). Moreover, the low number of photons per pixel during the brief pixel dwell time (effective dwell time down to 50 ns in 2D over time scanning) and the concomitant high Poissonian noise from the detection device (*Tian et al., 2012a*) further veil the behavior of individual ECC couplons. Microscopic characterization of individual ECC sites in the global population of ECC couplons is akin to the observation of an individual star in a galaxy. We thus considered employing the astronomical CLEAN algorithm (*Högbom, 1974*) to enable mapping of active ECC couplon sites, gain novel insights into the mechanism of cardiac ECC, and present a unique approach to mapping active ECC couplons in intact cardiomyocytes with unprecedented accuracy in the absence of any pharmacological intervention. Herein, we provide a fundamental analysis of the behavior of these ECC couplons and ECC reliability during electrically evoked $Ca^{2+}$ transients. We also describe their behavior during the negative $Ca^{2+}$ amplitude-frequency relationship and during β-adrenergic stimulation. Moreover, for the first time, we present a full 3D mapping of firing couplons during normal ECC.

## Results

### ECC couplon mapping and reliability of firing in mouse atrial myocytes

As previously reported, mouse atrial myocytes display a complex pattern of $Ca^{2+}$ signals during normal ECC (*Figure 1Aa*) (*Mackenzie et al., 2001*). Ultrafast confocal imaging (147 frames/second) allows for the monitoring of such complex $Ca^{2+}$ signals. Nevertheless, despite the chance to capture individual time points that display apparent initiation sites of local $Ca^{2+}$ signals (i.e., firing ECC couplons; see *Figure 1Aa* – panel 13.7 ms), these are solely 'eager sites' (*Mackenzie et al., 2001*), whereas ECC couplons firing later disappear in the evolving global $Ca^{2+}$ signal (see panels in *Figure 1Aa* later time points). Robust de-noising with a specialized 'CANDLE' (*Coupé et al., 2012*), algorithm, which simultaneously removes Poissonian and Gaussian noise, did not significantly improve this situation (*Figure 1Ab*). To improve on this scheme, we designed the CaCLEAN algorithm combining the astronomer´s CLEAN approach and known approximated diffusion properties of $Ca^{2+}$ ions (for a detailed description see Materials and methods section and *Figure 1—figure supplement 1*). The CaCLEAN method enabled us to identify active ECC couplons in each of the recorded frames (*Figure 1B*; for display reasons, we applied a slight blurring to the data; for the original CaCLEAN data, please see *Figure 1—figure supplement 2*). When we rebuilt the fluorescence images based on these active ECC couplon maps (*Figure 1C*), they resembled the original fluorescence very closely (compare *Figure 1A*). The Materials and methods section provides a detailed description of this rebuilding process.

To verify the arrangement of identified firing couplon sites with respect to their proximity to the adjacent plasma membrane, we imaged the distribution of the plasma membrane in the same cell after staining with CellMask DeepRed. We overlaid the cumulative active couplon map (*Figure 1Da*) and the plasma membrane in the same confocal section (*Figure 1Db*) and observed high similarity between identified release sites and the plasma membrane (*Figure 1Dc*, the structural similarity index (SSIM) (*Wang et al., 2004*) was 0.56 with an average of 0.57 ± 0.03, n = 46 from three animals). Because for CaCLEAN, we only considered the upstroke period of $Ca^{2+}$ transients, it was unimportant to capture the 'true' peak of the $Ca^{2+}$ transient, and contraction of the cardiac myocyte did not affect the determination of the ECC couplon sites (for additional experiments concerning that important aspect, see *Figure 1—figure supplement 3*).

We then investigated the degree to which $Ca^{2+}$ signals from 'out of focus' couplons might contribute to the ECC site maps generated by our CaCLEAN algorithm (z-axis resolution). To this end, we performed model calculations of the putative fluorescence signal from an individual firing couplon and varied its distance to the plane of focus (see *Figure 1—figure supplement 4A* and top row in B). We added recording noise (second row in *Figure 1—figure supplement 4B*) and fed the data through our CaCLEAN approach; the resulting maps of ECC couplon activity are depicted in the bottom row of *Figure 1—figure supplement 4B*. The peak CaCLEAN signal displayed a decreasing magnitude with an increasing distance to the plane of focus, resulting in an apparent z-resolution of approximately 1 µm under our recording conditions (see *Figure 1—figure supplement 4C*).

In addition to the z-axis resolution, we addressed the xy-resolution in three different ways. First, we simulated two local $Ca^{2+}$ transients (*Figure 1—figure supplement 5Aand B*) with various distances between their centres. These events were convoluted with noise levels also observed in our original recordings and passed through the CaCLEAN algorithm. From these data, we concluded that the CaCLEAN algorithm is able to resolve local transients whose centres are 1 µm apart (*Figure 1—figure supplement 5C*). In rat ventricular myocytes, the mean distance between RyR clusters was reported to be 1.02 µm (*Soeller et al., 2007*). Second, we simulated a $Ca^{2+}$ transient with gridded $Ca^{2+}$ release (*Figure 1*; *Figure 1—figure supplement 6Aa and b*). After adding realistic recording noise (*Figure 1—figure supplement 6Ac*), these data were passed through the CaCLEAN algorithm and couplon maps were generated (*Figure 1—figure supplement 6Ad*). Superimposing the calculated maps with the original simulate maps (*Figure 1—figure supplement 6B*) demonstrated a good correlation between these two data sets. Finally, we generated a single $Ca^{2+}$ release image with a descriptive model, as shown in the Materials and methods section, and distributed 1000 copies of such events randomly inside the circumvent of a typical ventricular myocyte (*Figure 1—figure supplement 7A,B andC*). The data were passed through the CaCLEAN algorithm (*Figure 1—figure supplement 7D*). Superimposing the centres of $Ca^{2+}$ release with the CaCLEANed couplon map

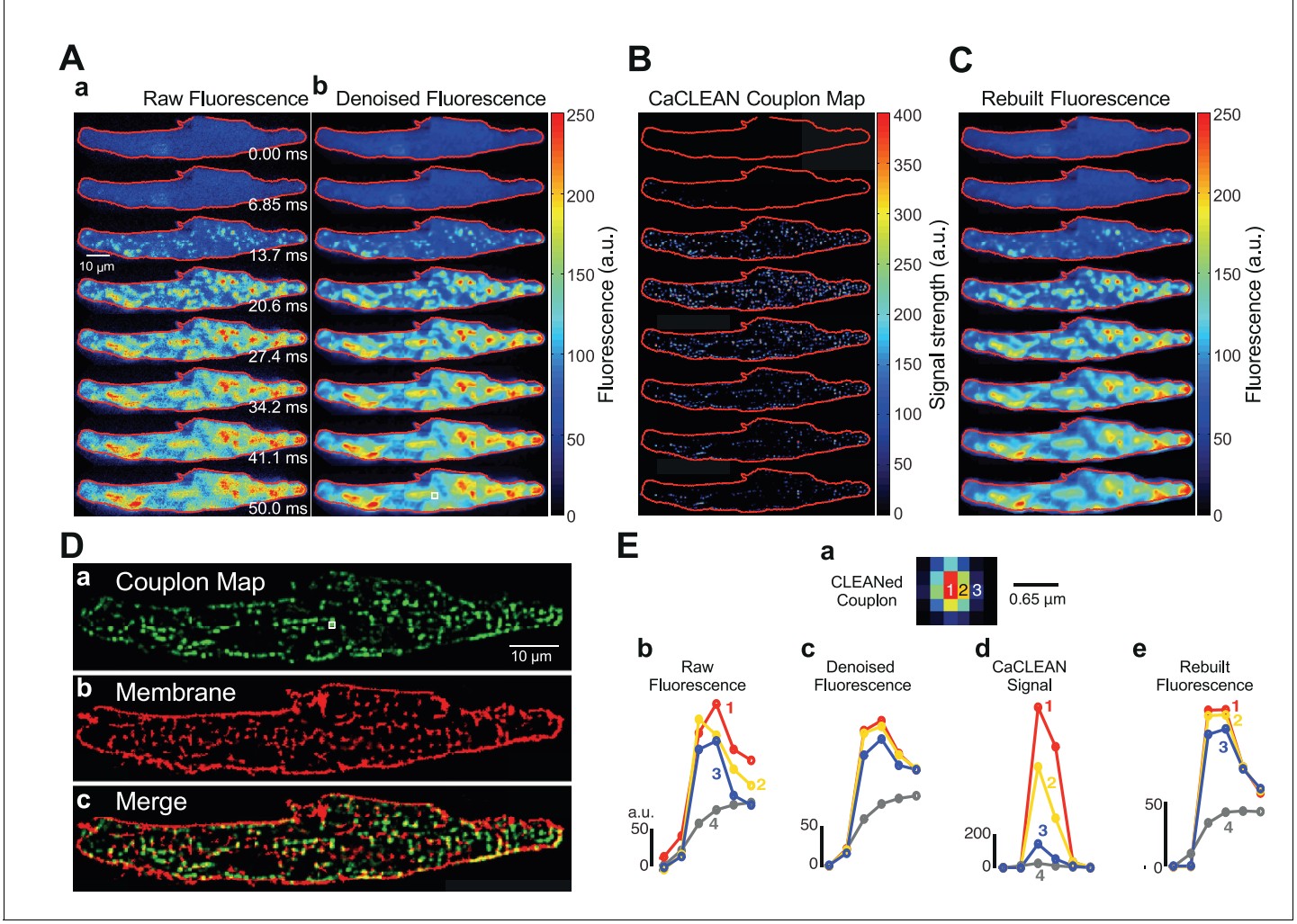

**Figure 1.** Identification, mapping and analysis of the behavior of ECC couplons in a mouse atrial myocyte during an individual $Ca^{2+}$ transient. (**A**) Panel a depicts raw confocal images of the first 50 ms of an electrically evoked $Ca^{2+}$ transient; panel **b** the denoised images from (**Aa**). (**B**) CaCLEANed images with $Ca^{2+}$ release sites clearly visible. (**C**) $Ca^{2+}$ transient rebuilt from the CaCLEANed image series in (**B**). (**D**) The couplon map (a) the plasma membrane topology (**b**) and an overlay of both (**c**). (**E**) Single pixel transients (traces 1~3) or global transient (trace 4) of the raw fluorescence (**b**) denoised fluorescence (**c**) CaCLEANed fluorescence (**d**) and rebuilt fluorescence (**e**), respectively. The positions of the single-pixel data are depicted in panel (**a**). Similar results were found in all cells analyzed (n = 46 myocytes from three animals).

DOI: https://doi.org/10.7554/eLife.30425.002

The following figure supplements are available for figure 1:

**Figure supplement 1.** Components of local $Ca^{2+}$ signals and the principle steps of the CaCLEAN algorithm.

DOI: https://doi.org/10.7554/eLife.30425.003

**Figure supplement 2.** Original, un-smoothed version of the CaCLEAN maps from *Figure 1B*.

DOI: https://doi.org/10.7554/eLife.30425.004

**Figure supplement 3.** Evaluation of potential contraction artifacts.

DOI: https://doi.org/10.7554/eLife.30425.005

**Figure supplement 4.** Behavior of the CaCLEAN algorithm with out-of-focus localized $Ca^{2+}$ transients.

DOI: https://doi.org/10.7554/eLife.30425.006

**Figure supplement 5.** Evaluation of the resolution limits of the CaCLEAN algorithm in the x-y axis.

DOI: https://doi.org/10.7554/eLife.30425.007

**Figure supplement 6.** Evaluation of the fidelity of the CaCLEAN algorithm with gridded $Ca^{2+}$ release sites.

DOI: https://doi.org/10.7554/eLife.30425.008

**Figure supplement 7.** Evaluation of the fidelity of the CaCLEAN algorithm with random $Ca^{2+}$ release sites.

DOI: https://doi.org/10.7554/eLife.30425.009

(*Figure 1—figure supplement 7E*) demonstrated that the CaCLEAN algorithm neither generated any false positive ECC couplons nor lost any $Ca^{2+}$ release events. On the basis of these tests in both the xy-plane and z-axis (*Figure 1—figure supplement 4*), we believe that the CaCLEAN algorithm can reproduce the physiological 3D distribution of active ECC couplons throughout rat ventricular myocytes.

Interestingly, the relationship between the CaCLEAN signal and the z-position was non-linear (see *Figure 1—figure supplement 4*), resulting in an augmented functional 'rejection' of 'out of focus' $Ca^{2+}$ signals and improved sectioning, as displayed in *Figure 1E*. We investigated the single-pixel $Ca^{2+}$ transients at the centre of an identified ECC couplon (pixel 1, red, *Figure 1Ea*), beside the centre pixel (pixel 2, yellow, 0.215 µm away from the central pixel, *Figure 1Ea*) and further away from the central pixel (pixel 3, blue, 0.43 µm away from the centre pixel, *Figure 1Ea*). Although the pixel $Ca^{2+}$ signals were comparable at all positions (*Figure 1Eb and C*), the resulting CaCLEAN transients displayed a very strong preference for the 'in focus' signals over those transients alongside the ECC couplon site (more than 10-fold difference in the peak intensity). Signals from positions located far from the couplon site were not considered (data not shown). Rebuilt fluorescence transients from the couplon map closely resembled the de-noised original data (compare *Figure 1Ec and e*), indicating that CaCLEAN identified all active couplon sites contributing to the cellular $Ca^{2+}$ transient.

Next, we further scrutinized the CaCLEAN approach. We first rebuilt three $Ca^{2+}$ transients from the same $Ca^{2+}$ transient (as depicted in *Figure 1Ab*) with variable Poissonian noise and Gaussian noise comparable to that observed during our recording, re-calculated their particular CaCLEAN maps as depicted in *Figure 2Aa-c,* and superimposed the three maps (*Figure 2Ad*). Because the three maps were RGB-color-coded, a perfect match between all three maps is indicated in white. As depicted in *Figure 2Ad and e*, we found a very good match (correlation coefficient = 0.84 ± 0.008; see *Figure 2Af*) between the three masks, supporting the high reproducibility of CaCLEAN calculations. Compared with the very good match between the active ECC couplon maps displayed in *Figure 2A*, the result was distinctly different when we analysed two consecutive $Ca^{2+}$ transients recorded under steady-state electrical pacing conditions (labelled rCaT1 and rCaT2 in *Figure 2Ba and b*). In the superimposed cellular overview (*Figure 2Bc*), as well as in the detailed magnified panels (*Figure 2Bd*, rightmost panel), it became obvious that the pattern of active ECC couplons was rather different between two consecutive $Ca^{2+}$ transients (average correlation coefficient 0.47 ± 0.025; see *Figure 2Bf*) despite superimposable global $Ca^{2+}$ transients (*Figure 2Be*). We identified 342 active ECC couplons in the first transient and 352 in the second but only 215 that participated in both $Ca^{2+}$ transients, indicating a re-fire rate of only 62.8%. We further superimposed active ECC couplon maps from consecutive $Ca^{2+}$ transients and observed a surprisingly low re-firing rate between consecutive $Ca^{2+}$ transients (*Figure 2—figure supplement 1A*, merge of three $Ca^{2+}$ transients). The firing probability of ECC couplons across four consecutive $Ca^{2+}$ transients demonstrated that only approximately 10% of couplons were always active (*Figure 2—figure supplement 1B*). These reliable couplons were probably the eager sites described previously (*Mackenzie et al., 2001*). Similar observations were made for all cells analyzed in this manner (n = 46).

## CaCLEAN-based quantitation of the ECC couplon firing reliability under physiological stimulation conditions

It remains unclear whether changes in the reliability of ECC couplon firing contribute to changes in cellular $Ca^{2+}$ signalling under physiological stimulation conditions, such as the negative transient-frequency relationship and/or the positive ionotropic effect of β-adrenergic stimulation. To address this relationship, we performed a series of experiments employing rat ventricular myocytes for which both phenomena are well established (*Dorn, 2010*). In cardiac myocytes, physiological augmentation of global $Ca^{2+}$ transients, for example following β-adrenergic stimulation, are envisaged to occur by various mechanisms, including modulation of the SR-$Ca^{2+}$ load, LTCC-mediated $Ca^{2+}$ influx or the number of recruited ECC sites (*Bers, 2008*). Rat ventricular myocytes display a well-known negative force-frequency relationship (*Anderson et al., 1973*), resulting, in part, from a decreased amplitude of the global $Ca^{2+}$ transient with increasing stimulation frequency (*Sipido et al., 1998*; *Gattoni et al., 2016*). We thus recorded and characterized confocal $Ca^{2+}$ transients with stimulation intervals of between 600 ms and 100 ms in the absence and presence of 10 nM

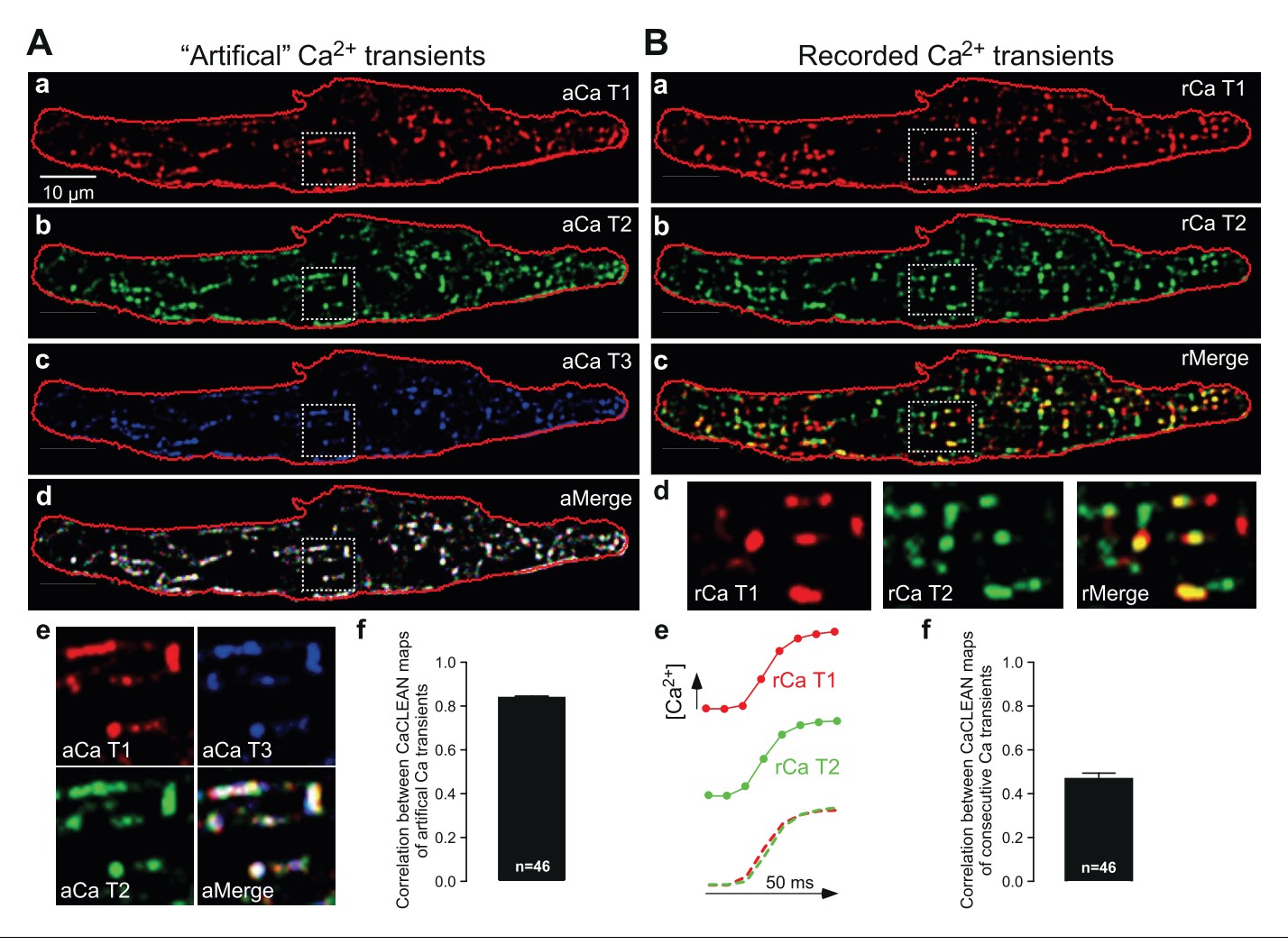

**Figure 2.** Robustness of CaCLEAN and reliability of ECC couplon firing.  (**A**) A denoised Ca²⁺ transient from the cell in *Figure 1* was convoluted with simulated noise as observed in our raw data. Three different artificial Ca²⁺ transients were generated and the resulting couplon maps were calculated with CaCLEAN (**a–c**, aCaT1–3). The maps were superimposed (**d** aMerge). Selected subcellular regions marked with dashed white boxes were replotted at higher magnification (**e**). (**f**) Statistical summary of the correlation coefficient between CaCLEAN maps of simulated Ca²⁺ transients. (**B**) Two consecutive electrically evoked Ca²⁺ transients from a mouse atrial myocyte were recorded in steady-state (@ 1 Hz stimulation frequency), the corresponding ECC couplon maps were constructed with CaCLEAN (**a** and **b**, rCaT1–2) and superimposed (**c**, rMerge). Subcellular regions marked with white dashed boxes (**a–c**) were replotted at a higher magnification (**d**). Panel (**e**) depicts individual global Ca²⁺ transients for CaT1 (red), CaT2 (green) and their overlay (from left to right). (**f**) Statistical summary of the correlation coefficient between CaCLEAN maps of consecutive Ca²⁺ transients in a population of cells.

DOI: https://doi.org/10.7554/eLife.30425.010

The following figure supplement is available for figure 2:

**Figure supplement 1.** Firing probability of ECC couplons in mouse atrial myocyte.

DOI: https://doi.org/10.7554/eLife.30425.011

isoproterenol (*Figure 3A and B*, respectively). The resulting global Ca²⁺ transients displayed the expected negative amplitude-frequency relationship, as depicted in *Figure 3Ca*. Isoproterenol substantially increased the Ca²⁺ transient amplitudes, particularly at low stimulation frequencies. The CaCLEAN algorithm generated ECC couplon maps for individual Ca²⁺ transients at stimulation intervals of 600 ms and 150 ms, as depicted in *Figure 3A* for control conditions and in *Figure 3B* in the presence of 10 nM isoproterenol. Isoproterenol promoted ECC under both the low and high stimulation frequencies (intervals of 600 ms and 150 ms, respectively), albeit to a vastly different degree. These ECC couplon maps were further segmented (watershed segmentation, see Materials and

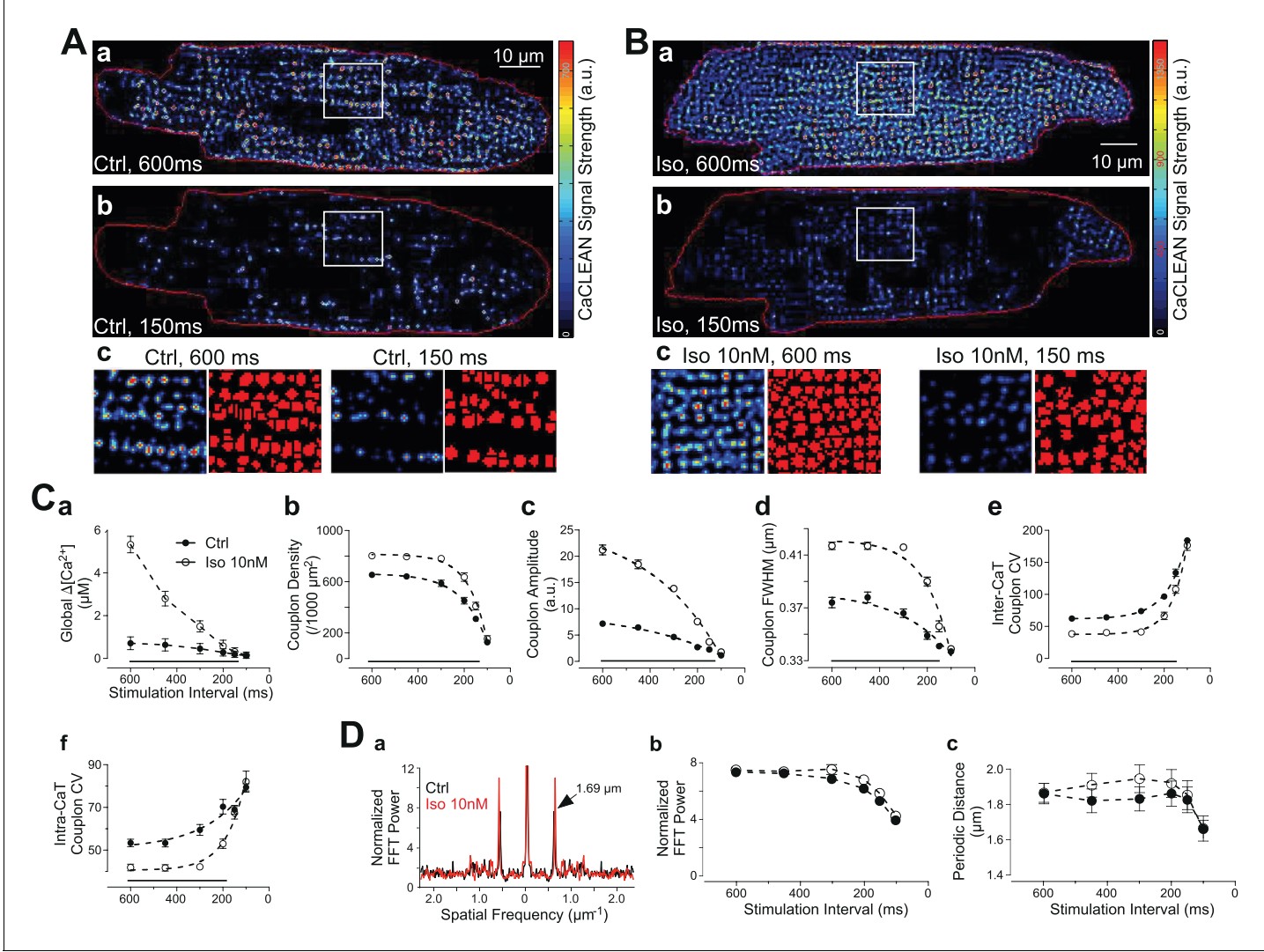

**Figure 3.** Behavior of ECC couplons at variable stimulation frequencies and during β-adrenergic stimulation. Rat ventricular myocytes were electrically paced into steady state at the stimulation intervals given. (**A**) The resulting CaCLEAN ECC couplon maps at 600 ms (**a**) and 150 ms (**b**) stimulation intervals under control conditions (Ctrl). Subcellular regions highlighted with white boxes were replotted at a high magnification (**c**) alongside with segmented ECC couplon sites (**c**, red). (**B**) Data similar to those in (**A**) but in a ventricular myocyte following β-adrenergic stimulation (10 nM isoproterenol, 5 min) (Iso). (**C**) (a) The negative $Ca^{2+}$ transient amplitude-stimulation frequency relationship for control conditions and following stimulation with 10 nM isoproterenol, here, global refers to the average of the confocal section (**b**) The density of active ECC couplons. (c) The released $Ca^{2+}$ amplitudes from individual ECC couplons and (**d**) the size of these ECC couplons (indicated as full width at half maximum, FWHM) decreases with increasing stimulation frequencies for control conditions (filled symbols) and following isoproterenol stimulation (open symbols); (e) the coefficient of variance (CV) of active ECC couplons increases between consecutive inter-CaT couplons and (**f**) within individual $Ca^{2+}$ transients (intra-CaT couplon CV) with increasing stimulation frequencies under control conditions and during β-adrenergic stimulation. Significant differences are indicated by the black lines in the lower parts of the panels. (**D**) Fast Fourier Transform (FFT) analysis of CaCLEAN maps at increasing stimulation frequencies under control conditions and following β-adrenergic stimulation. Panel (**a**) exemplifies the power spectra of the FFT analysis; panel (**b**) displays the normalized amplitudes of the characteristic power peaks (marked with an arrow in **a**) and panel (**c**) shows the characteristic spatial distances of those power peaks. For all conditions, at least 23 cells from three different animals were analyzed. Error bars smaller than their symbols were omitted. For statistical details see *Figure 3—source data 1*.

DOI: https://doi.org/10.7554/eLife.30425.012

The following source data and figure supplements are available for figure 3:

**Source data 1.** Detailed statistical summary of the data comparisons in *Figure 3CandD*.
DOI: https://doi.org/10.7554/eLife.30425.015

**Figure supplement 1.** CaCLEAN map (left part of each panel) and segmented ECC couplons (right part of each panel) for a typical rat ventricular myocyte under control conditions (**A**) and during simulation with 10 nM isoproterenol (**B**) at the given stimulation intervals.

*Figure 3 continued on next page*

*Figure 3 continued*

DOI: https://doi.org/10.7554/eLife.30425.013

**Figure supplement 2.** Comparison of CaCLEAN couplon map and anti-RYR immunolabeling with Fast Fourier Transform (FFT) analysis.

DOI: https://doi.org/10.7554/eLife.30425.014

methods for more details), as depicted in the typical magnified areas of the maps (red in *Figure 3Ac and Bc*). The complete set of CaCLEAN maps and segmented ECC couplon maps is summarized in *Figure 3—figure supplement 1*.

At each stimulation frequency, six consecutive, electrically evoked $Ca^{2+}$ transients were recorded under steady-state conditions. The resulting average densities of active ECC couplons of the control myocytes were $653 \pm 148$ per 1000 $\mu m^2$ (electrical stimulation @ 1.7 Hz; 600 ms intervals) and $310 \pm 146$ (electrical stimulation @ 6.7 Hz; 150 ms intervals) (*Figure 3Cb*, filled symbols). Isoproterenol increased the density of active ECC couplons ($803 \pm 35$ per 1000 $\mu m^2$ [@1.7 Hz] and $409 \pm 185$ [@6.7 Hz; *Figure 3Cb*, open symbols]). Under both conditions, the density of recruited ECC couplons dropped monotonically, albeit with a different quantitative relationship ($\tau_{ctrl}$=107 ms vs. $\tau_{iso}$=80 ms). In addition to the changes in ECC couplon density, isoproterenol promoted the released $Ca^{2+}$ content of each active couplon at low stimulation frequencies by more than 3-fold (*Figure 3Cc*). The full width at half maximum (FWHM) of these couplons also displayed small increases following isoproterenol stimulation (*Figure 3Cd*). Both the $Ca^{2+}$ release amplitudes of individual couplons and their FWHM also followed similar monotonic decay patterns with increasing stimulation frequency.

In a detailed quantitative investigation, we analysed the temporal behavior of ECC couplon firing across multiple $Ca^{2+}$ transients (inter-CaT, *Figure 3Ce*) and the spatial properties within the same $Ca^{2+}$ transient (intra-CaT, *Figure 3Cf*). We interpret the inter-CaT and intra-CaT coefficients of variation (CV) as the results of the firing reliability of ECC couplons because smaller values for CV result from 'more homogeneous' active ECC couplon maps. Thus, increases in CV are indicative of decreased reliability. Our data clearly demonstrated a decreasing firing reliability with increases in the stimulation frequency. For the low, physiological stimulation frequencies, application of 10 nM isoproterenol was able to increase temporal and spatial reliability significantly (*Figure 3Ce and f*). Interestingly, again at the highest pacing frequency, β-adrenergic stimulation was unable to increase reliability. These ECC couplon maps closely resembled the expected distribution of ECC couplons in healthy ventricular myocytes with periodic distances of 1.7 μm, as verified by a Fast Fourier Transform (FFT) analysis (*Figure 3Da*). The FFT power of these periodic distances in the couplon maps is comparable to the immunostaining results of RyRs (*Figure 3—figure supplement 2*). These data also indicate that our analysis is not strongly influenced by 'out-of-focus' $Ca^{2+}$ sources that would disturb and weaken the periodicity, that is, the power of the spatial frequency. Interestingly, the power of the characteristic spatial frequency decreases with decreasing stimulation interval (*Figure 3Db*), most probably because the failure rate of ECC couplons is increasing, while the value for the characteristic periodic distance was decreasing at the highest stimulation frequency, probably because of substantial increases in the diastolic $Ca^{2+}$ concentration (*Figure 3Dc*). It was noteworthy that neither of the two latter relationships was altered in the presence of 10 nM isoproterenol (open symbols in *Figure 3Db and C*).

## 3D reconstruction and functional mapping of active ECC couplons in electrically paced ventricular myocytes

The abovementioned results demonstrate our ability to map and reconstruct the distribution of active ECC couplons in naive living cardiac myocytes in two dimensions without pharmacological intervention. In the final series of experiments, we considered extending this approach in space and generating a three-dimensional (3D) map of active ECC couplons to reconstruct their spatial distribution in an intact ventricular myocyte. To this end, we followed the principle protocol detailed in *Figure 4A*. For each plane, we recorded the upstroke of a single electrical stimulation, calculated the CaCLEAN couplon map (exemplified in panels 1–5 of *Figure 4A*) and surface rendered the stacked maps. A typical distribution of active couplons is depicted in *Figure 4B*, in which the cross-striation arrangement of couplon sites can be recognized easily. To emphasize

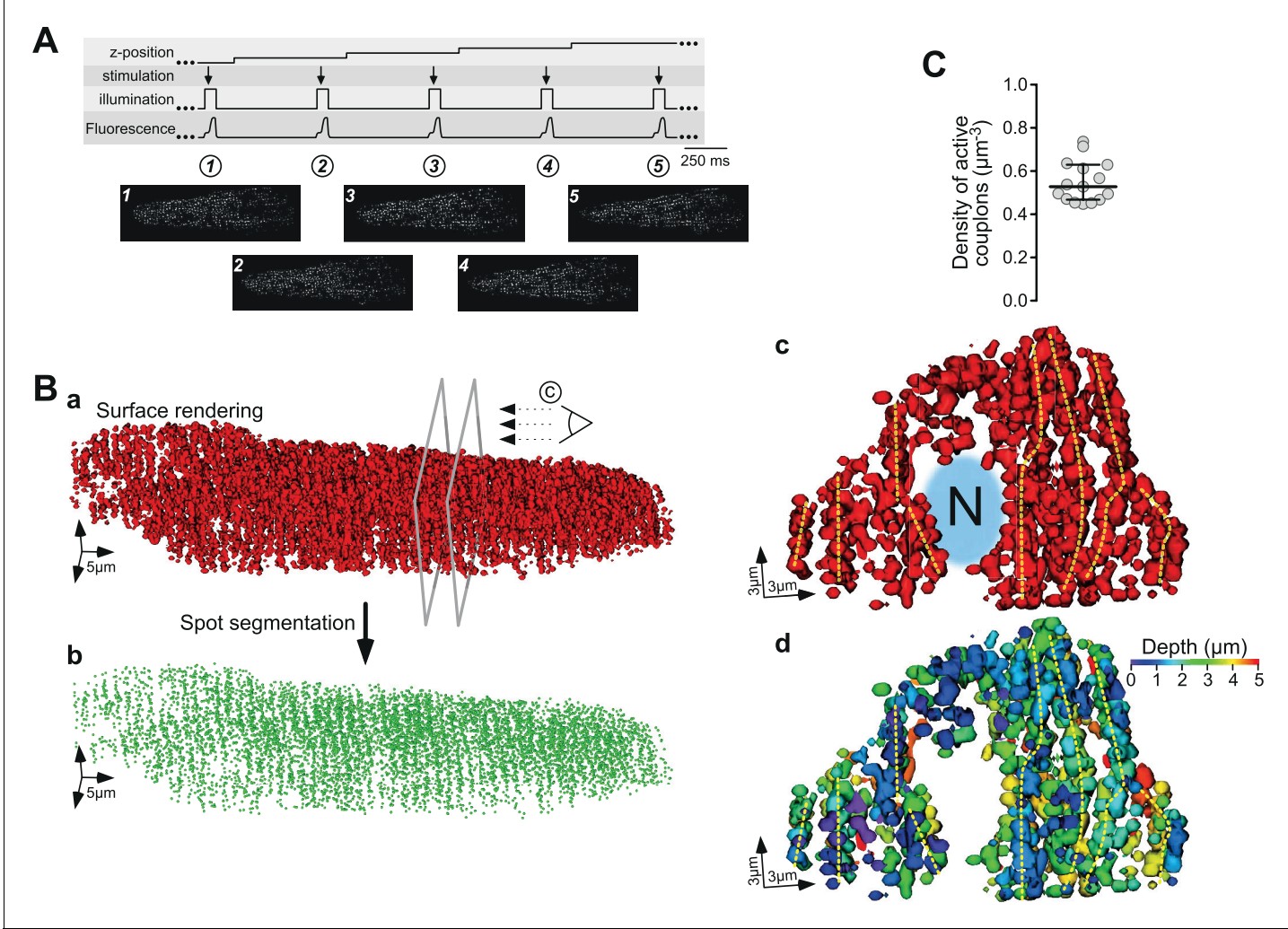

**Figure 4.** Three-dimensional reconstruction of active ECC couplons in a naïve rat ventricular myocyte. (**A**) Experimental regime for the 3D reconstruction of ECC couplon distribution. From top to bottom, the movement of the z-focus, the electrical field stimulation, the on and off status of the laser, and the upstroke of five consecutive $Ca^{2+}$ transients are displayed. The bottom panels depict exemplified consecutive CaCLEAN maps. (**B**) The resulting 3D surface rendering of active ECC couplons from a typical naïve rat ventricular myocyte (**a**) and their centers after spot segmentation (**b**); for panel (**c**), we cut the surface render open at the two planes indicated in (**Ba**) and turned the resulting object clock-wise into the paper plane. The stylized eye in (**Ba**) indicates the viewpoint ('N' and the blurred area in blue denote the nucleus). For (**Bd**) we color coded the depth of the individual objects with respect to the paper plane as indicated by the color wedge. (**C**) Summary of the mapping results as density of active ECC couplons calculated from 15 individual rat ventricular myocytes (from four hearts).

DOI: https://doi.org/10.7554/eLife.30425.016

The following figure supplement is available for figure 4:

**Figure supplement 1.** Still image a movie 1 illustrating the individual steps leading to 3D surface rendering.

DOI: https://doi.org/10.7554/eLife.30425.017

this feature, we performed a spot segmentation, as shown in *Figure 4Bb*. Here, the cross-striation of the ECC couplon arrangement is even more obvious. To further investigate the 3D arrangements of ECC couplon sites within the myocytes, we cut the rendered surface open at the two planes indicated by the grey frames and rotated the resulting surface model clockwise into the paper plane to observe the resulting object as indicated by the stylized eye in *Figure 4Ba*. *Video 1* illustrates the individual steps leading to the 3D surface rendering. The cuts were placed over one of the two typical nuclei of a ventricular myocyte, which can be recognized as the large free space in the middle, labelled 'N' in *Figure 4Bc*. A thorough analysis of the resulting surface-rendered representation indicates strings of couplons emanating deep into

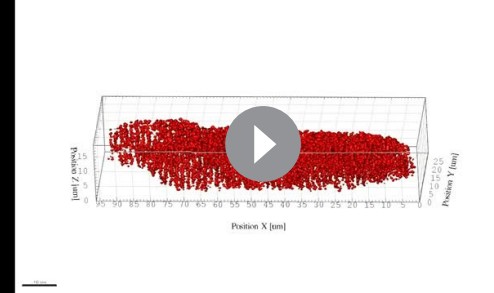

**Video 1.** Illustrating individual steps leading to 3D surface rendering of active ECC couplons.
DOI: https://doi.org/10.7554/eLife.30425.018

the inner space of the cell, as highlighted by the dashed yellow vertical lines in *Figure 4Bc*. We interpret those arrangements as couplons decorating T-tubular membrane systems. To facilitate three-dimensional visualization of this panel, we have redrawn the rendering and color-coded the depth of the individual ECC couplons according to the color wedge (*Figure 4Bd*). Such an experimental approach enabled us to estimate the total density of firing ECC couplons in a living ventricular myocyte. *Figure 4C* summarizes the results: on average, there were 0.55 couplons/$\mu m^3$. Given the typical volume of a rat ventricular myocyte of approximately 30,000 $\mu m^3$ (n = 15), the average number of active ECC couplons in a typical rat ventricular myocyte was approximately 16,500 couplons. This value correlates well with previously reported values (*Bers, 2008*).

## Discussion

The heart's pumping performance, measured in vivo as ejection fraction or fractional shortening, is substantially determined by the properties of the underlying cellular $Ca^{2+}$ transients. These $Ca^{2+}$ signals are the result of brief $Ca^{2+}$ influxes during the cellular action potential and of the subsequent triggered $Ca^{2+}$ release from the SR at sites of LTCC-RyR couplings, the so-called ECC couplons (*Soeller et al., 2007*). To date, detailed studies of ECC couplon behavior have been restricted to experimental conditions with strong $Ca^{2+}$ buffering (*Cleemann et al., 1998*; *Woo et al., 2002*), which are thought to uncouple individual ECC couplons during electrically evoked $Ca^{2+}$ transients, or with substantial inhibition of LTCCs, in which the number of firing ECC couplons is largely suppressed (*López-López et al., 1995*) (e.g., with verapamil, see *Figure 5*). Nevertheless, the detailed properties of such fundamental $Ca^{2+}$ signals have not been elucidated to date because of several major obstacles: (1) the speed of the $Ca^{2+}$ release process itself (in the millisecond range [*Cheng et al., 1993*]); (2) the close proximity of numerous ECC couplons (less than 1 $\mu m$ in a typical myocyte [*Hou et al., 2015*]); (3) the fast diffusion of $Ca^{2+}$ ions (in the range of tens of $\mu m^2$/s); and (4) the sheer amount of synchronized ECC couplons (many thousands in a typical myocyte) (*Bers, 2008*). Substantial noise during ultra-fast $Ca^{2+}$ imaging adds further complexity to the challenge of identifying active couplons and analyzing their behavior during electrically evoked cellular $Ca^{2+}$ transients.

In radio astronomy, researchers need to distinguish minute changes in the radio signal from a noisy background and determine the position of such radio sources with high precision. Because the intrinsic signal distribution is very difficult to derive with an interferometer, a thorough deconvolution has been employed instead (*Pety, 2007*). The CLEAN algorithm was introduced in the 1970s to perform such a deconvolution on images created in radio astronomy (*Högbom, 1974*). Briefly, the algorithm assumes that an image consists of a number of point sources for the radio signal. The algorithm iteratively finds the highest value in the image and subtracts a small representation of this point source convolved with the point spread function (called 'dirty beam' in radio astronomy) of the observatory until the radio signal has 'disappeared' below a set threshold (*Pety, 2007*).

The activity of individual ECC couplons in cardiac myocytes shares substantial similarity with the imaging process in radio astronomy: (1) the global $Ca^{2+}$ transient is the result of a large number of active ECC sites; (2) the underlying cardiac couplon serves as the point source; (3) the flux of $Ca^{2+}$ from these ECC couplons serves as the signal-emitting process; and (4) the $Ca^{2+}$ diffusion process can be readily approximated by the convolution of a light point source with the point spread function. On the basis of these analogies, we developed a novel and unique tool to investigate the behavior of individual ECC couplons in intact cardiomyocytes by combining our understanding of subcellular $Ca^{2+}$ diffusion and the classical CLEAN algorithm from radio astronomy. The results presented in this report strongly suggest that the proposed algorithm is remarkably powerful in

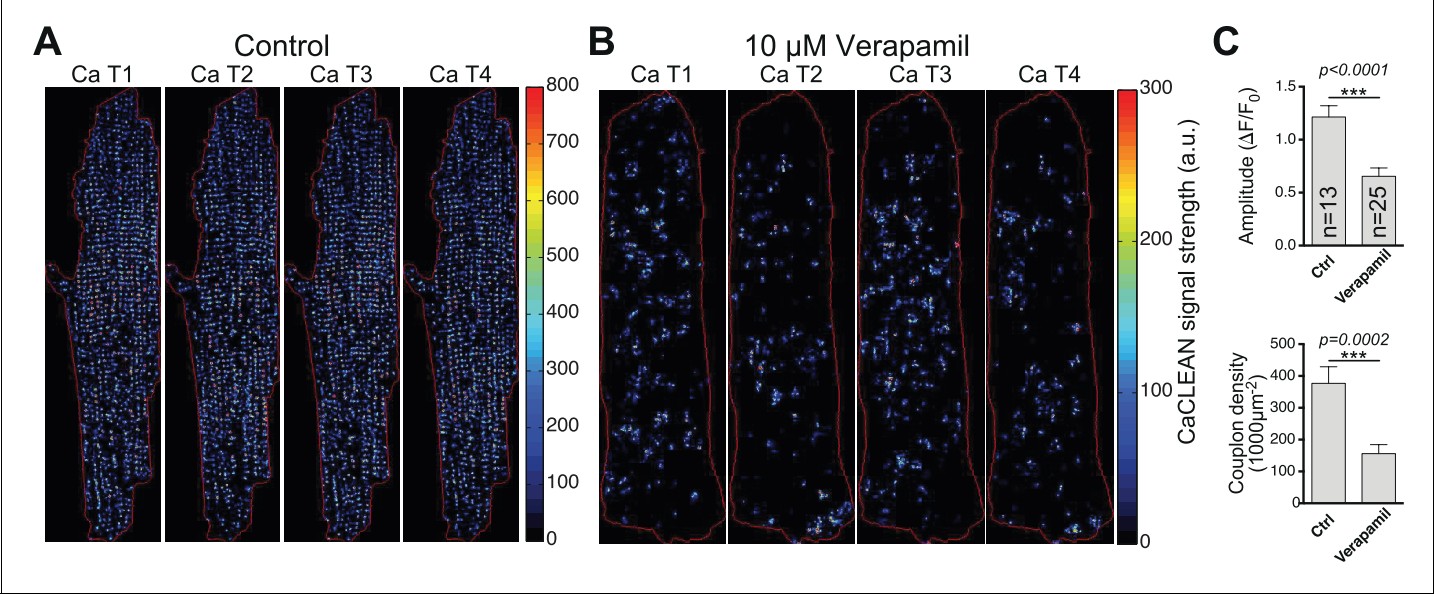

**Figure 5.** CaCLEAN-derived couplon distributions for typical rat ventricular myocytes (paced at 1 Hz). (**A**) Under control conditions and (**B**) in the presence of 10 µM Verapamil (10 min after application) to reduce the number of operating L-Type $Ca^{2+}$ channels substantially. Red lines outline the cell edges of the myocytes. (**C**) Statistical comparison of the 'global' $Ca^{2+}$ transient amplitudes (top panel) and the density of active couplons (bottom panel) for these two conditions. Please note that for this series of experiments, the external $Ca^{2+}$ concentration was lowered to 0.9 mM. Cells were analyzed from three different rats.

DOI: https://doi.org/10.7554/eLife.30425.019

deconvolving and identifying active ECC couplons and in determining minute differences in their resulting $Ca^{2+}$ signals and behaviors such as firing reliability.

Mouse atrial myocytes, with their complex and irregular membrane invaginations (*Kirk et al., 2003*), served as a test for evaluating CaCLEAN. ECC couplons are restricted to regions in close proximity to such membrane invaginations (*Kirk et al., 2003*). Indeed, an overlay of the CaCLEAN-derived couplon positions and the plasma membrane revealed strong structural similarity between these two data sets (see *Figure 1D*). Although $Ca^{2+}$ transients in the vicinity of couplon sites can still have substantial amplitude (*Figure 1Ea*), the CaCLEAN algorithm helps to discriminate positions at ECC couplon sites from those even a short distance away from such ECC centres (see *Figure 1Ec* and *Figure 1—figure supplements 4* and *5*). Thus, CaCLEAN also serves functionally as a 'deconvolution' algorithm. Moreover, the CaCLEAN algorithm displays substantial robustness because it generates virtually identical couplon maps from the same underlying $Ca^{2+}$ transient convoluted with random but different noise simulating our recording parameters (*Figure 2A*) and from simulated $Ca^{2+}$ transients (*Figure 1—figure supplement 6*).

To identify individual active ECC couplons, the CaCLEAN algorithm ought to provide spatial resolution in the x,y and z directions. As depicted in *Figure 6*, the two parameters for the $Ca^{2+}$-dye diffusion and the definition of ACO are rather critical. We determined the optimal values for these two parameters (60 $µm^2/s$ for $Ca^{2+}$-dye diffusion and 30 $µm^2/s$ for 'Analytical CLEAN Object' [ACO]) yielding the best resolution. For all three optical axis, we determined a resolution of 1 µm, a value that matched the average distance between putative ECC couplon sites well (*Soeller et al., 2007*). Several additional physical factors, such as the resolution of the objective, the pixel dimension of the camera and the binning used in the data recording, also contribute to the resolution of the CaCLEAN algorithm. Any manoeuvres that improve the signal-to-noise ratio of the recorded data, such as the use of $Ca^{2+}$ sensors with superior properties, will also help to improve the resolution of CaCLEAN.

Another approach to characterizing the function of couplon-mediated local $Ca^{2+}$ transients is employing high concentrations (1–5 mM) of a slow $Ca^{2+}$ buffer, such as EGTA. This method has been previously used in rat atrial (*Woo et al., 2002, 2005*) and ventricular myocytes

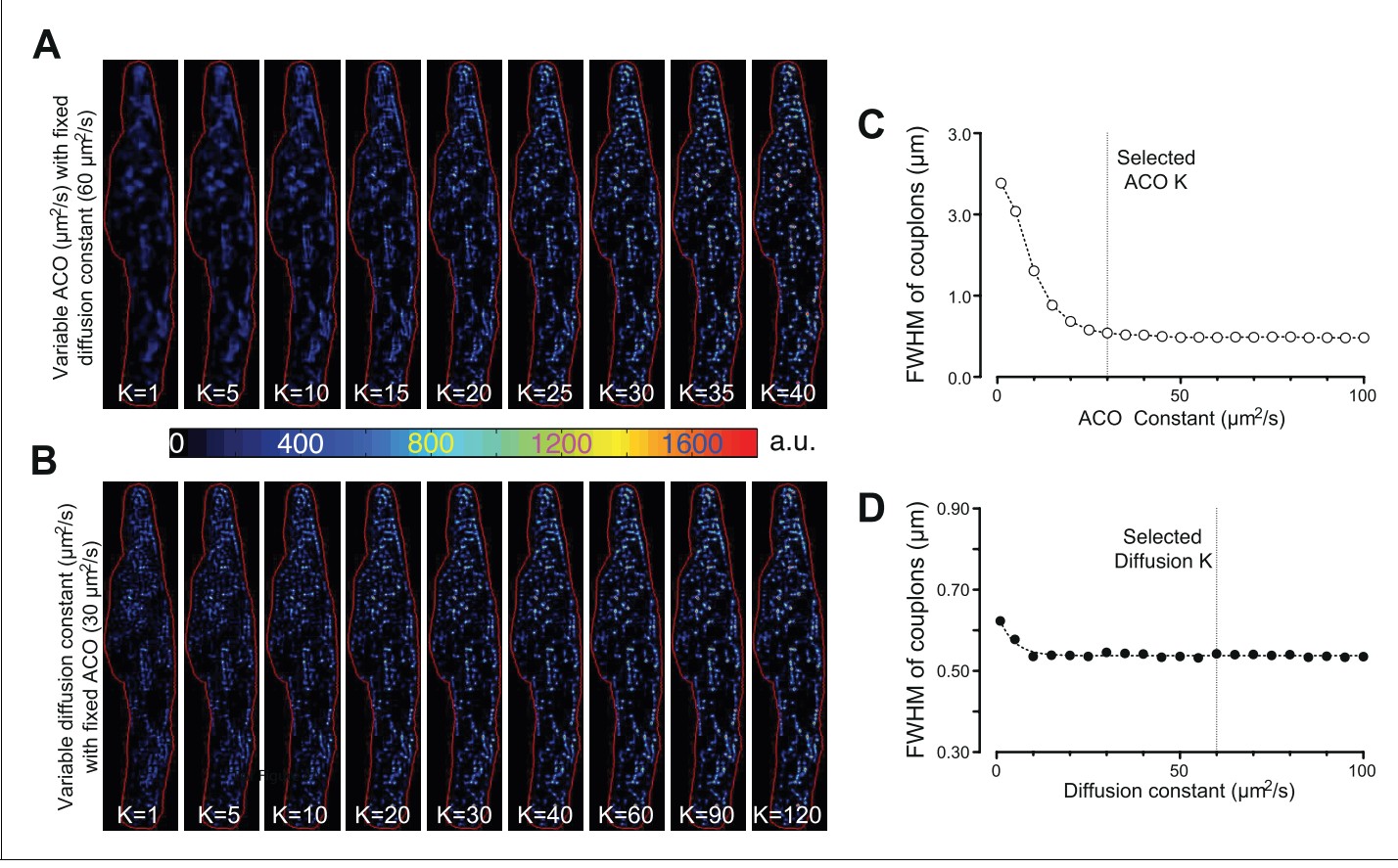

**Figure 6.** Effects of varying the ACO and diffusion constants on the characteristics of the detected couplon sites. (**A** and **B**) Couplon maps with variable ACO or diffusion constant, respectively. (**C** and **D**) Plots of full width at half maximum (FWHM) of the identified couplons (averaged over all identified couplons of the resulting map) against the variable ACO or diffusion constant, respectively. The dotted vertical lines indicate the values for the two constants chosen for the calculations in this report.

DOI: https://doi.org/10.7554/eLife.30425.020

(*Zahradníková et al., 2007*; *Janiek et al., 2012*) to identify and investigate the properties of ECC couplon sites. Our results in atrial myocytes (*Figure 1Dc*) are in line with previous findings indicating that ECC is restricted to subplasmalemmal regions (*Woo et al., 2002*, *2005*; *Mackenzie et al., 2001*). The report by *Woo et al. (2005)* in rat ventricular myocytes went a step further and compared their experimental results to data obtained by modeling the behavior of individual RyR clusters during ECC. This strategy allowed for a molecular interpretation of the measured 'Ca²⁺ spikes' (*Janiek et al., 2012*). Our results from rat ventricular myocytes (*Figure 3Ce and f*) also depicted the described variability of ECC in ventricular myocytes and extends these findings to the negative amplitude-frequency relationship in small rodents. Nevertheless, these approaches relied on patch clamping the cells and using excessive concentrations of a slow $Ca^{2+}$ buffer that might alter L-type $Ca^{2+}$ channels and/or RyR gating. By contrast, CaCLEAN is able to identify ECC couplons in intact ventricular myocytes with a low concentration of a fast $Ca^{2+}$ buffer, the $Ca^{2+}$ sensor.

The CaCLEAN algorithm assumes both symmetrical $Ca^{2+}$-dye diffusion and ACO; thus, barrier structures such as mitochondria or the SR itself might result in asymmetrical $Ca^{2+}$ diffusion. The symmetrical ACO does not represent a substantial problem because it is used as an almost nanoscopic component for de-composing the $Ca^{2+}$ signal in a highly iterative manner ($10^{-4}$ of the amplitude of the local $Ca^{2+}$ transient), but the initial steps in CaCLEAN are challenged by this assumption (see *Figure 1—figure supplement 1*). Here, the algorithm calculates a theoretical diffusion pattern based on the previous fluorescence distribution (assuming symmetrical diffusion). The severity of this simplified description of the diffusion process depends on (i) the temporal resolution and (ii) the

magnitude of asymmetry. Because an a priori estimation of inhomogeneous diffusion is essentially impossible, a high temporal resolution appears to be the only way to minimize this effect.

We were particularly surprised to find that despite indistinguishable global $Ca^{2+}$ transients, the subcellular patterns of firing ECC couplons were substantially different between individual global $Ca^{2+}$ transients (*Figure 2B*). Therefore, we posited whether the modulation of the reliability might be an important contributor to variations in the global $Ca^{2+}$ signal. In rat cardiac myocytes, the negative force-frequency relationship is established (*Gattoni et al., 2016*), and it was speculated that a number of processes contribute to this negative relationship, including modulation of the $Na^+/Ca^{2+}$ exchanger activity (*Yao et al., 1998*) and the SR $Ca^{2+}$ content (*Maier et al., 2000*; *Endoh, 2004*). We found that concomitant to the decreasing amplitude of the global $Ca^{2+}$ transient (*Figure 3Ca*, filled symbols), the density of participating ECC couplons substantially decreased with increasing stimulation frequency, as did the amplitude and size of the resulting local $Ca^{2+}$ transients (see *Figure 3Cc and d*). At stimulation frequencies above 10 Hz, the density of active ECC couplons dropped to virtually zero, indicating that the ECC mechanism was no longer able to follow such high pacing frequencies. Interestingly, we found in our studies that this cut-off frequency or stimulation interval coincided well with the time of RyR refractoriness (*Poláková et al., 2015*). We were able to identify the reliability of ECC couplons that participate in the global $Ca^{2+}$ transient as a contributor to the negative $Ca^{2+}$ transient amplitude-frequency relationship (*Figure 3Ce and f*). The coefficients of variance for both spatial and temporal reliability were increased two- to four-fold. These findings indicated that spatially synchronized recruitment of ECC couplons following electrical stimulation, an important prerequisite for optimized contractile output of the individual myocyte, was substantially de-synchronized or failing with increasing stimulation frequency. In addition, the reliability of ECC couplon recruitment from transient to transient faded as the pacing increased. From these findings, we conclude that the failure of spatially synchronous couplon recruitment and recruitment failures between transients are indeed substantial contributors to the ever-decreasing amplitude of global $Ca^{2+}$ transients with increasing stimulation frequency.

Interestingly, β-adrenergic stimulation not only caused more ECC couplons to contribute to the global $Ca^{2+}$ transient but also caused the global $Ca^{2+}$ transient to be more homogeneous (*Figure 3C*), and it is noteworthy that the reliability of consecutively recruiting ECC couplon sites from transient to transient was substantially increased. β-Adrenergic stimulation results in multiple phosphorylation events, including phosphorylation of LTCCs, phospholamban phosphorylation with the de-inhibition of the SERCA pump and direct RyR phosphorylation (*Bers, 2002*). Phosphorylation of LTCCs evokes an increased $Ca^{2+}$ trigger for RyR-mediated $Ca^{2+}$ release (*Catterall, 2015*; *Reuter, 1983*), whereas phospholamban phosphorylation leads to an increased SR-$Ca^{2+}$ load that fosters RyR sensitivity towards cytosolic $Ca^{2+}$ (*Tada and Toyofuku, 1998*). Moreover, direct RyR phosphorylation also renders the RyRs more sensitive to cytosolic and/or trigger $Ca^{2+}$ (*Marx et al., 2000*). Ultimately, all of these processes work in synergy and increase the reliability and recruitment of formerly 'silent' couplons during β-adrenergic stimulation.

The three-dimensional distributions of RyRs and LTCCs have been studied extensively (e.g., *Hou et al., 2015*), but to our knowledge, we present the first data on functional 3D maps of active, firing ECC couplons in living ventricular myocytes. We believe that such 3D functional mapping of ECC couplons in naive cardiac myocytes under steady-state stimulation conditions represents a milestone, and will enable further studies investigating the degree and mechanisms of distribution changes under physiological conditions. Furthermore, these findings can be the basis for further advancements in our understanding of structural and functional changes or maladaptation of ECC during very early states of cardiac diseases in which minute alterations in the RyR–LTCC relationship occur (*Gómez et al., 1997*). It is accepted that most cardiac diseases are accompanied by deterioration of ECC, amongst other mechanisms, by T-tubular remodeling (*Wei et al., 2010*). Our functional 3D mapping of active ECC couplons demonstrated strings of coupling sites, most likely alongside T-tubular membrane structures (*Figure 4Bc and d*), and thus represents a promising step toward studying the relationship between such functional structures in greater detail.

The CaCLEAN algorithm fuses the techniques of classical radio astronomy to knowledge about the biophysics of the cardiac ECC process to empower the identification, mapping and characterization of fundamental ECC coupling sites (couplons) in individual $Ca^{2+}$ transients from intact cardiac myocytes without pharmacological intervention. As such, we have been able to construct, for the first time, 3D maps of the distribution of active ECC couplons for naïve cardiac myocytes.

Experimental results from such analyses will further advance our understanding of the physiological, as well as the pathophysiological, behavior of these essential building blocks and their contribution to clinical conditions such as cardiac arrhythmia. Moreover, our approach will also be applicable to $Ca^{2+}$ signalling in other cell types such as neurons, where the properties of fundamental $Ca^{2+}$ signals such as those at synapses appear to be instrumental to their cellular and network processing of information.

## Materials and methods

### Cell isolation

Isolation of mouse atrial and rat ventricular myocytes was performed as described previously (*Reil et al., 2010*; *Tian et al., 2012b*).

### Confocal imaging

For $Ca^{2+}$ dye loading of cardiomyocytes, 1 µl $Ca^{2+}$ Fluo-8/AM, from a stock solution (1 mM in DMSO complimented with 20% Pluronic F-127) was diluted in 1 ml tyrode solution (140 mM NaCl, 5.4 mM KCl, 1.8 mM $CaCl_2$, 1 mM $MgCl_2$, 10 mM HEPES, 10 mM glucose, pH 7.35). The cells were incubated with the dye solution for 30 min at room temperature and were allowed an additional 15 min in dye-free tyrode for de-esterification. In experiments requiring simultaneous staining of the plasma membrane, CellMask DeepRed (Thermo Scientific, Germany) was used (0.2 µl in 1 ml solution) to label the plasma membrane for 15~30 min at room temperature. Dye-loaded cells on coverslips were mounted in a custom-made chamber onto the stage of a motorized Nikon microscope Eclipse Ti (NIKON, Tokyo, Japan). A high-speed 2D-array confocal scanning module, VT-Infinity (Visitech International, Sunderland, UK), was coupled to the microscope. Solid-state laser lines at wavelengths of 488 nm and 633 nm (Cobolt AB, Solna, Sweden) were combined and guided into the confocal module via a light guide. A Hamamatsu ORCA-Flash 4.0 sCMOS camera (Hamamatsu Photonics, Hamamatsu City, Japan) was coupled to the confocal module, and the images were transferred to a computer via camera link adaptors. The scanning speed of the confocal module was set to 600 Hz, and 60% of the chip area of the camera was used at its highest readout speed, resulting in a recording speed of 147 images/s. The cells were imaged through a Nikon Plan Fluor 60X oil DIC objective (NA = 1.40, NIKON, Tokyo, Japan). The built-in dichroic mirror in the confocal scanning head VT-Infinity was a ZT488/640rpc mirror (Chroma Technology Corporation, Vermont, USA). Fluo-8 (500–620 nm) and DeepRed (650–750 nm) fluorescence outputs were collected. The camera operated in its 2 × 2 binning mode, yielding a pixel size of 215 nm. The cells were electrically stimulated with a MyoPacer (IonOptics, Westwood, MA, USA), and durations of 8~12 s were typically recorded if not stated otherwise. In the experiments with varying stimulation frequencies or in the 3D reconstruction experiments, an external USB-2008 device (National Instruments, Munich, Germany) with custom C code was employed to generate triggers for synchronizing both laser switching and electrical stimulation, such that only the upstroke phase of a given $Ca^{2+}$ transient (25 ms for baseline and 50 ms for the upstroke) was captured to minimize cellular photo damage. To ensure steady-state conditions, all recordings were preceded by at least 25 electrical stimulations without illumination. All experiments were performed at room temperature.

RyR immunocytochemistry was performed as described below. Imaging of the dye-coupled secondary antibody was performed with similar hardware configurations and software settings, except that the exposure time was slightly longer (10 ms).

### Three-dimensional reconstruction of active couplons in rat ventricular myocytes

To reconstruct the arrangement of active LTCC/RyR couplons, rat cardiomyocytes were electrically paced at 1.6 Hz. After establishing steady-state conditions, the upstroke phases of 60 $Ca^{2+}$ transients were recorded following the basic protocol outlined in *Figure 4A*. The laser was switched on immediately before electrical stimulation (3–5 images), ; the cell was then stimulated, the upstroke phase of the $Ca^{2+}$ transient was recorded (typically seven images) and the laser was switched off. After another 100 ms, the z-focus was shifted 417 nm upwards. The data from such recordings were subjected to the CaCLEAN algorithm. Resulting maps of active couplons were

collected, combined and exported as a tiff file for further processing. 3D reconstructions were performed in Imaris software (Imaris 8.4.1; Bitplane AG, Zurich, Switzerland).

## Immunostaining of rat ventricular myocytes with anti-RyR antibody

Isolated rat ventricular cardiomyocytes were fixed with 4% PFA diluted in phosphate-buffered saline (PBS) solution (NaCl 137 mM, KCl 2.7 mM, $Na_2HPO_4$ 10 mM, $KH_2PO_4$ 1.8 mM, pH 7.4) for 20 min and washed three times with PBS. Cells were permeabilized with 1% TritonX 100 (diluted in PBS) for 20 min, washed once with PBS, and blocked with 5% BSA in PBS for 30 min at room temperature. Following BSA blockage, the cells were incubated with mouse anti-mouse RyR2 IgG polyclonal antibody (1:100, Thermo Scientific) at 4°C overnight, washed three times for 10 min with 0.1% Tween diluted in PBS (PBS-T) and then incubated with goat anti-mouse DyLight 649 antibody (1:300, Jackson Immunoresearch Labs Inc.) for 1 hr at room temperature. Finally, cells were washed three times for 10 min with PBS-T and mounted with ProLong Gold antifade reagent (Invitrogen, Germany) for further imaging.

## The Ca²⁺ CLEAN algorithm

### The components of the upstroke signal from a cardiac Ca²⁺ transient

Consider two consecutive image frames (time point $t_0$ and $t_1$, respectively, with the time interval $\Delta t$, $\Delta t = t_1 - t_0$) during the upstroke phase of a $Ca^{2+}$ transient; the corresponding fluorescence signals indicating $Ca^{2+}$ concentrations are $F_{t_0}$ and $F_{t_1}$, respectively. The fluorescence at $t_1$ comprises three parts—the diffused fluorescence from $F_{t_0}$ ($F_{t_0|\Delta t}^{Diffusion}$), the newly increased fluorescence ($F_{\Delta t}^{Release}$) and the removed fluorescence by $Ca^{2+}$ ion transporters ($F_{\Delta t}^{Removal}$)—and can be formulated as follows:

$$F_{t_1} = F_{t_0|\Delta t}^{Diffusion} + F_{\Delta t}^{Release} - F_{\Delta t}^{Removal} \tag{1}$$

The fluorescence from $t_0$ will diffuse during the time interval $\Delta t$, which should be thoroughly considered,

$$F_{t_0} = \iint_{-\infty}^{+\infty} F_{t_1}^{diffused}(x,y) dx dy \tag{2}$$

where $F_{t_1}^{diffused}(x,y)$ denotes the fluorescence level at time point $t_1$ and at a specific two-dimensional spatial position $(x,y)$ derived from the fluorescence $F_{t_0}$. The two-dimensional diffusion equation has a numerical solution at time point $t_1$ (after $\Delta t$, where $\Delta t > 0$) for a single point source (*Patankar, 1980*):

$$\Phi_{(\Delta t,x,y)} = \frac{F_{t_0}}{4\pi D \Delta t} e^{-\frac{x^2+y^2}{4\pi D \Delta t}} = F_{t_0|\Delta t}^{Diffusion} = F_{t_1}^{diffused} \tag{3}$$

where $D$ denotes the diffusion constant of $Ca^{2+}$-dye fluorescence. Although this is not an exact mathematical solution for an entire image comprising thousands of point sources, it is assumed to be a good approximation.

From time point $t_0$ to $t_1$, there are two major sources that contribute to the fast increase in $F_{\Delta t}^{Increase}$: fluorescence of the portion of $Ca^{2+}$ ions from the extracellular environment via LTCCs and fluorescence of $Ca^{2+}$ released from the SR via RyRs. For cardiac couplons, $Ca^{2+}$ influx via LTCCs and $Ca^{2+}$ release via RyRs are spatially very close to each other (separated by a few nm) and cannot be distinguished optically; therefore, they can be considered as the same $Ca^{2+}$ release source/site ($F_{couplon|\Delta t}^{Release}$). Thus, we have

$$F_{\Delta t}^{Increase} = F_{LTCC|\Delta t}^{Release} + F_{RyR|\Delta t}^{Release} = F_{couplon|\Delta t}^{Release} \tag{4}$$

The measured fluorescence of $Ca^{2+}$ will be counterbalanced by the removal of $Ca^{2+}$ by four major transport entities: (1) the SERCA, (2) the $Na^+/Ca^{2+}$ exchanger (NCX), (3) sarcolemmal $Ca^{2+}$-ATPase, and (4) the mitochondrial $Ca^{2+}$ uniporter system.

$$F_{\Delta t}^{Removal} = F_{SERCA|\Delta t}^{Removal} + F_{NCX|\Delta t}^{Removal} + F_{Ca^{2+}-ATPase|\Delta t}^{Removal} + F_{Mito-Ca^{2+}-Uniport|\Delta t}^{Removal} \tag{5}$$

Thus, *Equation 1* has the following solution:

$$F_{t_1} = \frac{F_{t_0}}{4\pi D\Delta t} e^{-\frac{x^2+y^2}{4\pi D\Delta t}} + F^{Release}_{couplon|\Delta t} - \left( F^{Removal}_{SERCA|\Delta t} + F^{Removal}_{NCX|\Delta t} + F^{Removal}_{Ca^{2+}-ATPase|\Delta t} + F^{Removal}_{Mito-Ca^{2+}-Uniport|\Delta t} \right) \quad (6)$$

RyRs release $Ca^{2+}$ into the cytosol for only a brief period (approximately 10 ms), whereas the removal of $Ca^{2+}$ requires approximately 300 ms (*Cheng et al., 1993*). Considering the low $Ca^{2+}$-removal speed compared with that of $Ca^{2+}$ release via RyRs and LTCCs, as well as our goal of developing only an analytical method, the $Ca^{2+}$-removal processes ($F^{Removal}_{\Delta t}$) could be neglected when analyzing only the upstroke phase and the temporal resolution is sufficiently high, e.g., less than 10 ms/frame. On the basis of the abovementioned equations (*Equation 1, 3 and 4*), we simplified the fluorescence at $t_1$ to:

$$F_{t_1} = \frac{F_{t_0}}{4\pi D\Delta t} e^{-\frac{x^2+y^2}{4\pi D\Delta t}} + F^{Release}_{couplon|\Delta t} \quad (7)$$

This simplified formalism (*Equation 7*) enables us to investigate the behavior of ECC couplons by analysing the $Ca^{2+}$ transient that can be imaged by ultrahigh-speed confocal microscopy (*Tian et al., 2012a*). The simplified process of $Ca^{2+}$ increase is depicted in *Figure 1—figure supplement 1A*.

## Separation of $Ca^{2+}$ diffusion and $Ca^{2+}$ release

Before the data can be processed by the CLEAN algorithm, the two main contributors of the measured fluorescence, namely, diffusion of the fluorescence from the previous time point ($\frac{F_{t_0}}{4\pi D\Delta t} e^{-\frac{x^2+y^2}{4\pi D\Delta t}}$) and additional fluorescence from newly released $Ca^{2+}$ ions ($F^{Release}_{couplon|\Delta t}$), ought to be separated. Throughout this text, we deliberately refer to 'release' alone, although we appreciate that $Ca^{2+}$ influx and $Ca^{2+}$ release both contribute to the couplon-derived $Ca^{2+}$ signal. We also deliberately simplify and approximate our considerations of the 2D conditions in our plane of focus, although we are aware that diffusion into and out of our plane of focus is indeed a 3D problem. To the best of our knowledge, we have checked, verified and scrutinized our 2D considerations and would refer the reader to *Figure 1—figure supplement 4* and *Figure 3—figure supplement 2* (and accompanying text). First, an empirical diffusion function of $Ca^{2+}$ and $Ca^{2+}$-dye molecule complexes was calculated with the apparent averaged cytosolic diffusion constant for $Ca^{2+}/Ca^{2+}$-dye diffusion (see *Figure 6*). It is noteworthy that the spread of the ECC couplons (as indicated by the full width at half maximum, FWHM) displayed little dependence on the apparent diffusion constant K (see *Figure 6*). We thus selected a value (60 μm²/s) that was approximately in the middle of the published range (15–120 μm²/s, *Safford and Bassingthwaighte, 1977*; *Nakatani et al., 2002*), as indicated by the vertical dotted line in *Figure 6*. Therefore, we calculated a theoretical residual 'image' based on the distribution of the preceding image and subtracted it from the actual distribution, as illustrated in the left panel of *Figure 1—figure supplement 1B*. This operation was repeated for all images in an image series and resulted in a stack containing images with the isolated $Ca^{2+}$ release signal.

## The CLEAN process

The CLEAN algorithm was inspired by an algorithm developed in the late 1970s in the field of radio astronomy to extract real structures from sidelobe disturbances (*Högbom, 1974*). In this study, a small 'Analytical CLEAN Object' (ACO) was generated on the basis of the $Ca^{2+}$-dye diffusion constant ($K$) and the time interval ($\Delta t$) for every frame:

$$ACO = \frac{1}{4\pi K\Delta t} e^{\frac{x^2+y^2}{4\pi K\Delta t}} = \frac{1}{4\pi \left(\frac{D}{2}\right)\Delta t} e^{-\frac{x^2+y^2}{4\pi \left(\frac{D}{2}\right)\Delta t}} \quad (8)$$

which was in principle similar to the diffusion function but with a smaller (50%) diffusion constant ($\frac{D}{2}$, 30 μm²/s) and with a very small intensity, usually one millionth of that of the recorded signal. We tested the effect of the variation of the detected couplon site from the ACO constant thoroughly, as shown in *Figure 6*. For every 'pure' $Ca^{2+}$ release image, the highest pixel was identified and used as the initial centre of the CLEAN ACO. Then, a copy of the CLEAN ACO was subtracted from the calculated $Ca^{2+}$ release signal, and the position of the CLEAN ACO was recorded (*Figure 1—figure supplement 1B*, right panel). This process was repeated until no further pixel above a threshold was found (approximately one million times). This threshold was determined

empirically to yield no false-positive events (see below). All obtained CLEAN ACO centre positions were superimposed and used as the final position of the couplon site. The entire CLEAN process is summarized in the right panel of *Figure 1—figure supplement 1B*.

## The rebuild process of the Ca$^{2+}$ transient

The rebuilding algorithm within CaCLEAN essentially represents the reverse process of the CLEAN approach. The algorithm used the ECC couplon maps from the CaCLEAN algorithm to reconstruct the upstroke phase of the Ca$^{2+}$ transient. We employed the resulting data to scrutinize the accuracy of the CaCLEAN algorithm. Briefly, the rebuilding process comprised two components: 1) the residual from the previous frame of the Ca$^{2+}$ transient, which was calculated with the Ca$^{2+}$-dye diffusion constant ($D$); and 2) the diffusion of the newly release Ca$^{2+}$ within the current frame, which was calculated by convoluting the counting information (including the spatial information and the number of steps) of the CLEAN process with the ACO itself. In addition, to account for the function of the SERCA pump, 1.5% of both the residual and newly released Ca$^{2+}$ was subtracted from the current frame ($F = F - F \cdot rate \cdot t$, where $rate = 0.00219/ms$). This rebuilding process was repeated for each frame during the upstroke phase of a given Ca$^{2+}$ transient.

## Implementation of the CaCLEAN algorithm

The entire CaCLEAN algorithm was implemented in MatLab (The MathWorks, Inc., Natick, MA, USA). For faster processing, CaCLEAN's kernel loop was programmed in C and interfaced to Mat-Lab/MEX. For better visualization, the images with superimposed centres were slightly smoothed with a GCV-based algorithm (using a smooth parameter s = 0.3) (*Garcia, 2010*). Imaging data were imported into MatLab R2014a with custom code based on the LOCI bio-format tools (*Linkert et al., 2010*). The appropriate short image series carrying the upstroke phase of each resulting Ca$^{2+}$ transient was extracted with custom code in MatLab. The resulting stacks data were de-noised with custom MatLab functions based on the 'CANDLE' de-noising algorithm (*Coupé et al., 2012*). A reference image containing the camera offset and the basal fluorescence was calculated by averaging two or three frames (14 or 21 ms) prior to electrical stimulation. The image was subtracted from all subsequent frames. Finally, the preprocessed data were passed to the proposed CaCLEAN algorithm. After CaCLEANing, the results containing meta-data were automatically saved in the '.mat' file format. The results can be simultaneously exported in the 'tiff' format. All these procedures were integrated and programmed into a single command in MatLab that ran in batch mode without user intervention. Before the execution of the program, all parameters were set by a graphical user interface-based dialog.

The resulting couplon maps were segmented with a built-in watershed algorithm in MatLab. The segmented ECC couplon masks were further refined with a cell mask, and the regions below 5% of the local maximum (in the surrounding 1.5 µm) were rejected. Such segmentation of release sites was performed on every CaCLEAN map. For the experiments involving rat ventricular myocytes, six consecutive Ca$^{2+}$ transients were used to calculate the coefficient of variation (CV) between successive Ca$^{2+}$ transients (referred to as temporal CV; *Figure 3Cc*). The CV of couplons within the same Ca$^{2+}$ transient was calculated from all segmented couplon sites within the same Ca$^{2+}$ transient of the cell (referred to as spatial CV; *Figure 3Cd*). All these procedures were compiled into one command in MatLab that ran in batch mode without user intervention.

## Determination of the constants for ACO and Ca$^{2+}$ diffusion

The optimal values for the two important constants, ACO and Ca$^{2+}$-dye diffusion, were determined empirically (*Figure 6*). A typical Ca$^{2+}$ transient from a mouse atrial myocyte was subjected to the CaCLEAN algorithm, either with varying constants for ACO while keeping the diffusion constant fixed (60 µm$^2$/s, *Figure 6A&C*) or with varying constants for the Ca$^{2+}$-dye diffusion while keeping the ACO constant fixed (30 µm$^2$/s, *Figure 6B&D*). The resulting CaCLEANed maps of active couplons were then segmented as described above. The centre of gravity of each segmented couplon was calculated, and then the distances of all pixels to the gravity centre were calculated. The intensities of each pixel were plotted against the calculated distances and then fitted to a Gaussian function $G_{(x,a,s)} = a * e^{-\frac{x^2}{2s^2}}$, where 'a' indicates the central height of the couplon, x the distance and s the half width of the Gaussian function. The full width at half maximum (FWHM) was calculated as two

times the parameter s. The selected values are highlighted by the vertical dotted lines in *Figure 6C and D*.

## Determination of an optimal threshold for CaCLEAN

The threshold of the CaCLEAN process determines the break conditions of the iterative core (see *Figure 1—figure supplement 1*; labelled 'Until No Maximum Found') and is thus essential for avoiding detection of false-positive couplon sites. To determine the minimal threshold for avoiding false-positive events, we recorded the basal fluorescence of a typical rat ventricular myocyte (stack with 500 images) and averaged the data series to obtain a virtually 'noise-free' image. This image was duplicated seven times yielding an image series similar to those necessary for an upstroke of a typical $Ca^{2+}$ transient. The resultant image stack was convoluted by various detector gain levels (*Bers, 2002*; *Wier and Balke, 1999*; *Lipp and Niggli, 1994*; *Hou et al., 2015*; *Catterall, 2015*) while keeping the white noise constant ($\sigma = 5.95$). These simulated recordings were finally passed to the CaCLEAN algorithm, and different thresholds were evaluated for each simulated recording detector. For typical background signal-to-noise levels (bgrSNR; approximately 4), the results are summarized in *Figure 7*. From these results, it is apparent that a minimum threshold of 10 safely avoided detecting false-positive events.

## Evaluation and simulation of noise

The measured raw fluorescence intensity is proportional to the photon count in each individual pixel:

$$I = P_x \cdot \alpha + N_{(\delta, \sigma^2)} \qquad (9)$$

where $\alpha$ denotes the detector gain, $P_x$ denotes an average photon flow of x that follows Poisson distribution, and $N_{(\delta, \sigma^2)}$ denotes white noise with a standard deviation of $\sigma$ and a detector offset of $\delta$. Gaussian noise and the gain value of the camera were evaluated with the PureDenoise plugin

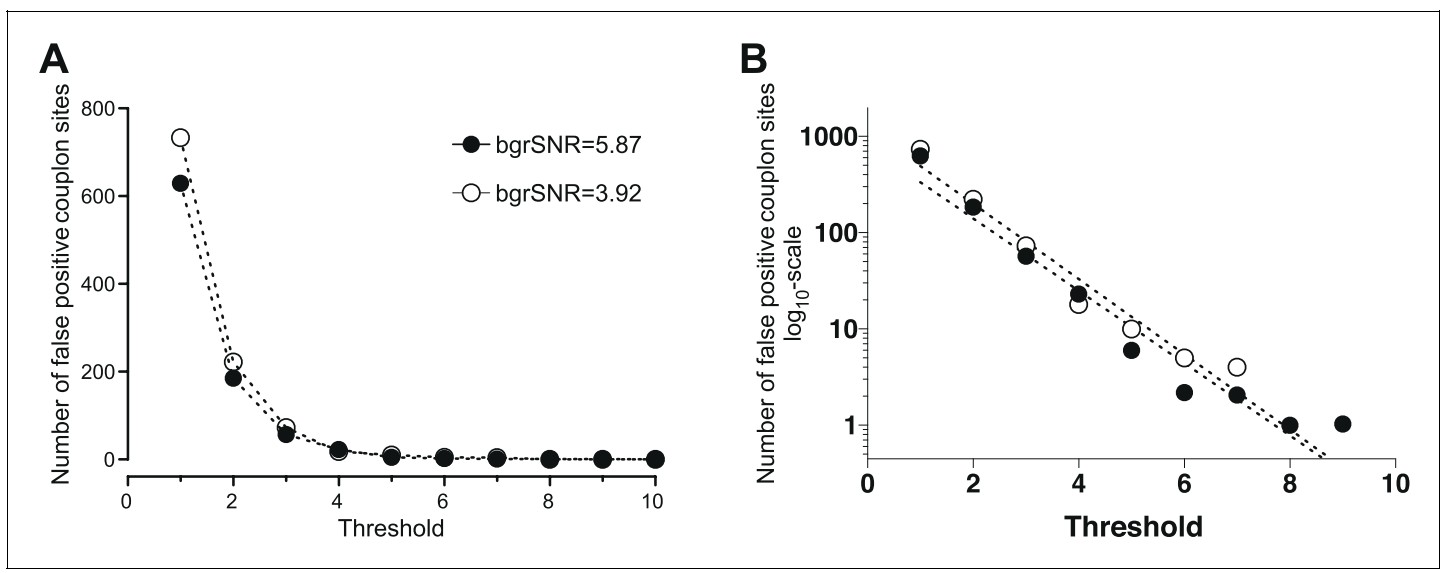

**Figure 7.** Determination of the minimal threshold for the CaCLEAN algorithm for avoiding false-positive events. We recorded the basal fluorescence of a typical rat ventricular myocyte and averaged the data series to obtain a virtually 'noise-free' image. This image was duplicated seven times (equal to 48 ms) yielding an image series similar to those necessary for the CaCLEAN calculation during the upstroke phase of $Ca^{2+}$ transients. The resultant image stack was convoluted with various signal amplification levels of the detector (also known as detector gain levels, e.g. 1, 3, 10, 20, 30) while keeping the white noise constant ($\sigma = 5.95$). These simulated recordings were finally passed to the CaCLEAN algorithm assuming various threshold levels. (**A**) For background signal to noise ratios (bgrSNR) that were close to our raw recordings (around 4), the number of false-positive couplon sites (original simulated data did not contain ANY $Ca^{2+}$ release sites) are plotted against the assumed thresholds. (**B**) The same data as in (**A**), but this time the numbers of false-positive couplons have been plotted logarithmically and fitted to a linear function. We determined a threshold value of 10 as the optimal value to avoid false-positive detection of couplons while not discarding positive couplons.
DOI: https://doi.org/10.7554/eLife.30425.021

(http://bigwww.epfl.ch/algorithms/denoise/) (*Luisier et al., 2010*) in ImageJ (https://imagej.nih.gov/ij/index.html). Gaussian noise and Poissonian noise were simulated with the 'randn' and 'poissrnd' commands in MatLab R2014a, respectively.

## Simulation of local Ca$^{2+}$ transient

A descriptive model was used to simulate a local Ca$^{2+}$ transient at a putative ECC couplon site, based on properties known from elementary Ca$^{2+}$ release events (*Cheng and Lederer, 2008*), whose central pixel transient is described by

$$I_{(t)} = \frac{A}{2} e^{\frac{\sigma^2 + 2\tau(\mu - t)}{2\tau^2}} \left( 1 - \frac{\sigma^2 + \tau(\mu - t)}{|\sigma^2 + \tau(\mu - t)|} \cdot \mathrm{erf}\left( \frac{|\sigma^2 + \tau(\mu - t)|}{\sqrt{2}\sigma\tau} \right) \right) + B \tag{10}$$

where $A$ represents the amplitude, $B$ the basal fluorescence, and $\sigma$, $\tau$, and $\mu$ the full width at half maximum (FWHM), the transient's decay constant and the maximum time point of Ca$^{2+}$release, respectively. The FWHM changes of the Ca$^{2+}$ signal are described by:

$$\sigma_{(t)} = \begin{cases} \sigma_m \left( 1 - e^{-\frac{t-\mu}{\tau_s}} \right), & if \left( 1 - e^{-\frac{t-\mu}{\tau_s}} \right) \geq 0 \\ 0, & if \left( 1 - e^{-\frac{t-\mu}{\tau_s}} \right) < 0 \end{cases} \tag{11}$$

where $\tau_s$ denotes the FWHM change rate over time, $\mu$ the onset of the Ca$^{2+}$ transient, and $\sigma_m$ the maximum FWHM. A 2D Gaussian function was used to describe the spatial extent of the transient,

$$Local\ calcium_{(x,y,t)} = I_{(t)} * e^{-\frac{x^2+y^2}{2(\sigma_{(t)})^2}} \tag{12}$$

where $x$ and $y$ denote the spatial position of single pixels. By convolving the intensity function, the FWHM function and the 2D Gaussian function, it was possible to simulate local Ca$^{2+}$ transientsusing parameters obtained from the literature (the $\sigma_{upstroke}$ = 3 ms, $\tau_{decay}$ = 40 ms, FWHM$_{max}$ = 3 μm, $\tau_{FWHM}$ = 10 ms) (*Cheng and Lederer, 2008*; *Smith et al., 1998*).

## Simulation of local Ca$^{2+}$ transient with z-focus shifting

The simulated 2D over time local Ca$^{2+}$ signal (as described in the section 'Simulation of local Ca$^{2+}$ transient') was converted to a 3D over time Ca$^{2+}$ transient by rotating every frame from the x,y-plane to the z-dimension. The 3D over time local Ca$^{2+}$ transient was then resliced at variable distances from its spatial centre.

## Noise tolerance test of CaCLEAN algorithm

A typical Ca$^{2+}$ transient from a mouse atrial myocyte (shown in *Figure 1A*) was thoroughly investigated by employing the PureDenoise plugin in ImageJ as mentioned above. The standard deviation of the Gaussian noise was 5.93, and the device gain was estimated to be 2.7. The measured Ca$^{2+}$ transient was de-noised with the CANDLE algorithm (see above) and further referred to as the 'original data' transient (*Figure 8*). To vary the Poissonian noise component, the Gaussian noise was kept fixed, and the device gain was changed to 5, 10, 15 and 20, respectively. The real photon flux for each pixel was then calculated, and Poissonian noise was generated on the basis of the photon flux of each pixel. Subsequently, the resulting 'noisy' image stack was multiplied by the appropriate gain value to 'simulate' real recordings. The resultant fluorescence transients were then subjected to CaCLEAN (*Figure 8*). To evaluate the effect of varying Gaussian noise, the device gain was kept fixed, and Gaussian noise was generated with its standard deviation changed to 0, 10, 20, 30 and 40, respectively. The Gaussian noise was added to the 'original data' fluorescence transient, and the resultant image stack was analysed with CaCLEAN (*Figure 8C*). The results of this analysis are summarized in *Figure 8D*. As depicted, we could identify a drop in the number of identified couplons only for very low signal-to-noise ratios below four. All data recorded and analysed for this report were characterized by signal-to-noise values above 5 (indicated by the blue arrow in *Figure 8D*).

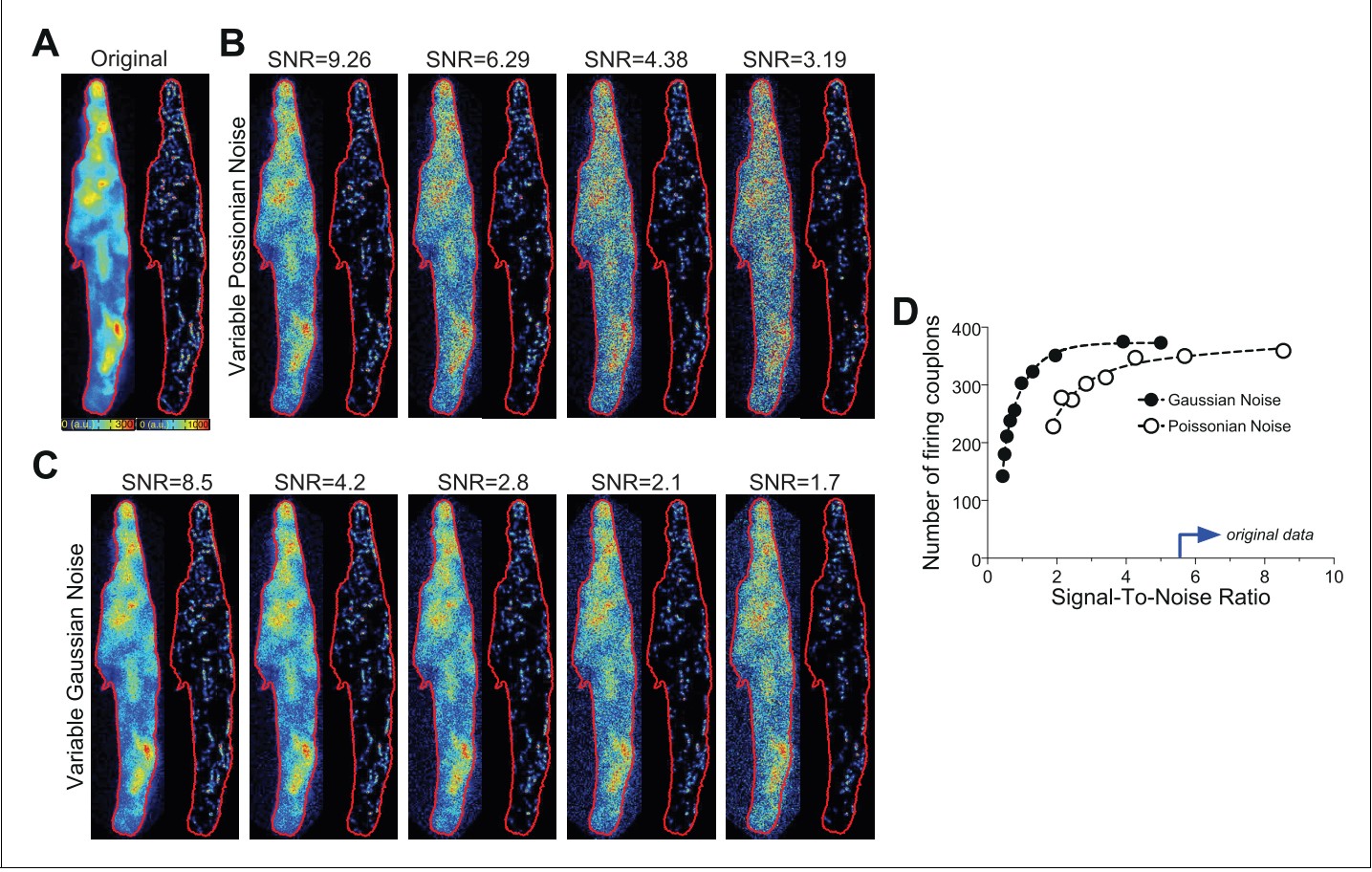

**Figure 8.** Noise tolerance test of the CaCLEAN algorithm. (**A**) Original fluorescence image (left) from the peak of an electrically evoked $Ca^{2+}$ transient in a mouse atrial myocyte and the resulting CaCLEAN couplon map (right). (**B** and **C**) Exemplified fluorescence data (left) and resulting CaCLEAN couplon maps (right) for various levels of added Poissonian (**B**) and Gaussian noise (**C**). (**D**) Summary of the entire data set. The blue arrow at the x-axis marks the lowest signal-to-noise ratio (SNR) in the data displayed in this report, emphasizing that the CaCLEAN algorithm can still preform robustly in even the 'worst' data sets.

DOI: https://doi.org/10.7554/eLife.30425.022

## Simulation of the upstroke phase of a $Ca^{2+}$ transient with gridded $Ca^{2+}$ release sites

First, a sample diastolic cell was loaded with Fluo-8 and recorded, as described in the confocal imaging section. The recorded data were used as the basal fluorescence. Second, a probability map matching the sample cell with gridded release sites was generated. For each release site, the probability was calculated with a 2D Gaussian function, $P = e^{-\frac{x^2+y^2}{2\sigma^2}}$, where $\sigma$ was set to 100 nm. The distances to neighbouring releases sites in the x- and y-axis were set to 1.075 μm (five pixels) and 1.935 μm (nine pixels), respectively. Then, 34,000 localized $Ca^{2+}$ transients were simulated as described in the 'Simulation of local $Ca^{2+}$ transient' section (the $\sigma_{upstroke}$ = 1 ms, $\tau_{decay}$ = 3 ms, $FWHM_{max}$ = 0.2 μm, $\tau_{FWHM}$ = 1 ms). These events were placed into the sample cell according to the probability maps, and the occurrence of these events temporally followed a Gaussian distribution $P = e^{-\frac{t^2}{2\sigma^2}}$, with $\sigma$ of 3 ms. Finally, the simulated $Ca^{2+}$ release transient was convoluted with a diffusion function calculated according to *Equation 3*. Noise was simulated as described in the section 'Noise tolerance test of CaCLEAN algorithm'.

## Simulation of the upstroke phase of a $Ca^{2+}$ transient with random $Ca^{2+}$ release sites

First, 1000 localized $Ca^{2+}$ transients were simulated as described in the 'Simulation of local $Ca^{2+}$ transient' section (the $\sigma_{upstroke}$ = 3 ms, $\tau_{decay}$ = 40 ms, $FWHM_{max}$ = 3 µm, $\tau_{FWHM}$ = 10 ms). These events were put into a sample cell with random x- and y-positions, and the occurrence of these events temporally followed a Gaussian distribution $P = e^{-\frac{t^2}{2\sigma^2}}$, with $\sigma$ of 6.85 ms. Finally, the simulated $Ca^{2+}$ release transient was convoluted with specific noise as described in the section 'Noise tolerance test of CaCLEAN algorithm'.

## Fast Fourier Transform analysis of the CaCLEAN maps

Two-dimensional CaCLEANed maps were transformed into the frequency field. The transformed images were then normalized with an average of their surrounding baselines with a distance of 1/3 $µm^{-1}$ and further smoothed with the GCV-based smoothing algorithm (s = 0.3) as mentioned above. The characteristic power peak (see for example, *Tian et al., 2012b*; *Hammer et al., 2010*) along the longitudinal axis of the myocytes was identified, and the maximum value of that peak and its horizontal position (i.e., spatial frequency) were extracted.

## Correlation or structural similarity analysis

Because the CaCLEAN maps and membrane staining were characteristically different, the traditional correlation coefficient was not suitable; thus, the structural similarity index (SSIM [*Wang et al., 2004*; *Rivenson et al., 2016*]) was used to test the similarity between CaCLEAN maps and the corresponding membrane staining. A similar approach was described recently to evaluate the registration of image structures (here, the plasma membrane and couplon sites) (*Saha et al., 2017*). The built-in function ssim in MatLab was used to calculate the similarity index by considering only the structural terms while omitting the luminance and contrast terms. For tests between two similar images, for example two ECC couplon maps from two consecutive $Ca^{2+}$ transients, traditional correlation coefficients were used and calculated with the MatLab built-in function *corr2*.

## Evaluating effects of contraction artefacts on the CaCLEAN algorithm

Because the increase in $Ca^{2+}$ is tightly coupled with cellular contraction, the CaCLEAN maps might be affected by the high $Ca^{2+}$ concentration that occurs later in the upstroke phase of $Ca^{2+}$ transients. The CaCLEAN maps were calculated on the basis of the upstroke phase of the electrically evoked $Ca^{2+}$ transients. We thus evaluated the degree to which contraction contributes to alterations in the CaCLEAN maps by changing the geometry of the cell. *Figure 1—figure supplement 3* summarizes the results of this analysis. On the basis of these findings, we concluded that because no detectable cell shortening occurred during the analysis time of the CaCLEAN algorithm, contraction was unlikely to alter or blur the CaCLEAN maps presented here.

## Statistical information and data presentation

Figure panels were generated in MatLab using custom-designed code. The line/symbol plots were prepared in GraphPad Prism 6 and 7 (GraphPad, La Jolla, CA, USA). All $Ca^{2+}$ imaging data were processed in their original arbitrary units (a.u.).

Experiments were repeated in at least three animals if not otherwise stated. The exact number is given in each figure legend, where 'n' indicates the total number of cells analysed. Cells were only analysed when they displayed a rod-shaped morphology and a lack of excessive spontaneous activity at the end of each experimental procedure, indicating their 'healthy' condition.

All data were initially tested for their normal distribution in GraphPad Prism 7.0 using a D'Agostino-Pearson omnibus normality test. In this case, the data were normally distributed pairs of data sets, which were compared using an unpaired t-test; otherwise, a Mann-Whitney test was employed. Statistical significance was indicated depending on the p-value: * for p<0.05, ** for p<0.01, *** for p<0.001 and **** for p<0.0001. Data are presented as mean values ± standard error of the mean (SEM). An outlier analysis was not performed. In line plots with data symbols, error bars were omitted in cases in which they were smaller than the symbols themselves. In line plots with data symbols, significantly different data pairs (tested as detailed above) were labelled with a black line at the

bottom and the significance level to facilitate reading of the figure (see, e.g., *Figure 3C and D*). To facilitate interpretation of *Figure 3C,E and F*, we simplified the representation of statistical significance. For a detailed summary of all statistical comparisons in *Figure 3C,E and F*, we refer the reader to *Figure 3—source data 1*.

In *Figure 3*, $Ca^{2+}$ concentrations were calculated using a mathematical approach described by *Cheng et al. (1993)*.

Source code is available at https://github.com/qhtian/CaClean published under a GPLv3 license (*Tian, 2017*). A copy is archived at https://github.com/elifesciences-publications/CaCLEAN.

## Acknowledgements

We are particularly grateful to Sabrina Hennig for her excellent technical support. This work was funded in parts by the DFG (SFB 894/TP A19 and TRR 152/TP P17) to PL and the Medical Faculty (HOMFORexcellence) to QT.

## Additional information

### Funding

| Funder | Grant reference number | Author |
| --- | --- | --- |
| Saarland University, Medical Faculty | HOMFORexcellence | Qinghai Tian<br>Jia Guo |
| Deutsche Forschungsgemeinschaft | SFB 894/TP A19 | Peter Lipp |
| Deutsche Forschungsgemeinschaft | TRR 152/TP P17 | Peter Lipp |

The funders had no role in study design, data collection and interpretation, or the decision to submit the work for publication.

### Author contributions

Qinghai Tian, Conceptualization, Data curation, Software, Formal analysis, Funding acquisition, Investigation, Visualization, Methodology, Writing—original draft, Writing—review and editing; Lars Kaestner, Conceptualization, Methodology; Laura Schröder, Data curation, Investigation; Jia Guo, Data curation, Formal analysis, Investigation; Peter Lipp, Conceptualization, Supervision, Funding acquisition, Validation, Investigation, Visualization, Methodology, Writing—original draft, Project administration, Writing—review and editing

### Author ORCIDs

Peter Lipp http://orcid.org/0000-0003-4728-9174

### Decision letter and Author response

Decision letter https://doi.org/10.7554/eLife.30425.026
Author response https://doi.org/10.7554/eLife.30425.027

## Additional files

### Supplementary files

• Transparent reporting form
DOI: https://doi.org/10.7554/eLife.30425.023

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
