## [Decision Letter]

Thank you for submitting your article "Fusion of Cardiology and Astronomy Empowers Characterization and Functional 3D Mapping of Individual Excitation Contraction Coupling Sites" for consideration by *eLife*. Your article has been favorably evaluated by Anna Akhmanova (Senior Editor) and three reviewers, one of whom is a member of our Board of Reviewing Editors. The following individual involved in review of your submission has agreed to reveal their identity: Michael Stern (Reviewer #3).

The reviewers have discussed the reviews with one another and the Reviewing Editor has drafted this decision to help you prepare a revised submission.

Summary:

The study by Tian and colleagues introduces and evaluates an interesting new approach to reconstruct the spatial and temporal distribution of Ca^2+^ release sites in living cells. The approach (CaCLEAN) uses the CLEAN algorithm of radio astronomy convolved with a Ca^2+^ diffusion model to localize and characterize Ca^2+^ release events in cardiac myocytes under normal excitation-contraction coupling (ECC) conditions.

The significance of the new technique is that it makes possible for the first time, analysis of triggered calcium signaling events at thousands of closely spaced ECC couplons in intact muscle, thus providing a new window on mechanisms of physiological ECC. The authors describe a variety of tests to validate the technique, and apply it to study the bases for the negative Ca^2+^ amplitude-frequency effect and the effects of β-adrenergic stimulation. By revealing the behavior of individual Ca^2+^ release units under different conditions, the technique has the potential to distinguish changes in local Ca^2+^ flux amplitude and the probability and reliability of localized Ca^2+^ release events. The technique provides a powerful new tool to study mechanisms underlying physiological and pathophysiological behavior of the heart, and may have broader application to a wide variety of calcium signaling events in other cell types as well.

Essential revisions:

All three reviewers agreed that the CaCLEAN technique is innovative and potentially powerful, and will interest a broad community of researchers in the area of calcium signaling. However, the reviewers found the current description and validation of CaCLEAN inadequate. As described the technique is semi-empirical, choices of key parameters are not experimentally constrained, and there are not enough data to evaluate the technique's accuracy in identifying the true locations of Ca^2+^ release events. To address these issues, the following essential concerns must be adequately addressed before the paper can be accepted.

1) CaCLEAN needs additional validation before it can be accepted as ground truth about calcium release events. The performance of the algorithm depends highly on underlying assumptions of parameters and the choice of diffusion coefficients. More confidence is needed that the location of release sites is an unbiased estimate of their true location, and that the presence or absence of a spark at a given location is not influenced by variations in dye loading, light intensity etc.

a) The resolution limits of the CaCLEAN technique and implications for interpretation of results need to be discussed more quantitatively. What proportion of the CaCLEAN sites are individual as opposed to overlapping couplons? Thresholding is used in the CLEAN iterations to avoid false positives, but how does the choice of threshold affect the number of missed events (i.e., couplons)?

One approach to give a more complete description of the accuracy and limits of the technique would be to simulate dye images of large numbers of closely localized simultaneous sparks at known locations, and determine how precisely and accurately those locations are recovered by the CaCLEAN process. Ideally this would involve a 3D reaction-diffusion model, but this may not be practical for a variety of reasons, including the time required, and the lack of consensus on a single calcium model with buffering for the myocyte.

As a reasonable compromise, one could generate a single spark image using existing spark models, and linearly superimpose a large number of copies with various centers to see if CaCLEAN can recover the centers. This would address the major question of whether clustering and overlapping of the fluorescence patterns cause any systematic bias in CaCLEAN, even though it would not take into account all the nonlinear buffering interactions among sources. This should be do-able, preferably with the sparks placed in 3 dimensions and then "imaged" with the confocal PSF.

In this way one could specify a range of critical conditions (couplon spacing, S/N ratio, assumed values for CLEAN thresholding and diffusion coefficients) required for the algorithm to generate an accurate map, and to place quantitative limits on its accuracy. Such information will be essential to validate the technique and to enable others to use it properly.

b) As a further consistency check, it would be useful to use the algorithm to reconstruct release sites in quiescent myocytes that exhibit spontaneous sparks. Raw frame data for that analysis should be readily available (perhaps from frames between stimulations). This test should allow a visual comparison between observed sparks and ECC release sites which would provide a consistency check as well as demonstrate the small subset of couplons that are engaged in spontaneous activity.

c) The algorithm assumes symmetrical diffusion, raising the question of how the reconstruction would be affected by diffusion barriers closed to couplons, e.g., mitochondria. Ideally this would be addressed by using appropriate simulated data, but if this is beyond the scope of the current study, a brief discussion of such potential effects should be included.

2) Better quantification in several places is needed.

a) Regarding the proximity of Ca^2+^ release sites to the plasma membrane in Figure 1, the text refers only to a "very good correlation." A quantitative measure should be included.

b) In Figure 2, the masks show a "very good match" while in Figure 2 the pattern of couplons is "rather different." Important information could be gleaned by measuring the probability of firing for each couplon over multiple stimuli, revealing the location of couplons that are highly reliable vs. those that are not. Are the most reliable couplons the first to fire? The number of cells analyzed should also be stated.

c) In Figure 1, Ca^2+^ signals are measured at different distances from entry sites, but they are not strictly comparable because they are recorded in the vicinity of different entry sites. It would be more informative to compare responses at variable distances from the same entry site, and to average several examples if possible.

3) The algorithm, and in particular the ACO, needs to be more clearly described. How is the ACO generated, what is the equation for it, and how are the particular choices of ACO parameters justified? Is the ACO amplitude or size the more important variable? How does the choice of ACO parameters (amplitude, diffusion coefficient) affect the lateral spatial resolution?

4) The application of CaCLEAN to understand the negative amplitude-frequency relationship is a potential strength of the paper. Early on the authors mention decreased SR Ca^2+^ content and a decreased number of participating couplons as possible causes, but their respective contributions are not fully resolved here. A decreased number of couplons certainly contributes (Figure 3Cb), but are the Ca^2+^ transient amplitudes of active couplons constant or do they decline with frequency? One way to address this would be to calculate the mean and s.d. of Ca^2+^ signal time course from the entire ensemble of couplons at low and high driving frequencies, and plot the average F vs. time to illustrate any changes. Multiplying this waveform by the number of active couplons may then demonstrate whether these changes are sufficient to account for the observed Ca-frequency relationship.

In the Abstract, the negative amplitude-frequency relationship is attributed to decreased firing reliability at higher frequencies. However, it is not clear whether release reliability (as defined by the temporal and spatial CV) tells us anything more than the decline in the total number of active couplons. This should be clarified.

---

## [Author Response]

Essential revisions:All three reviewers agreed that the CaCLEAN technique is innovative and potentially powerful, and will interest a broad community of researchers in the area of calcium signaling. However, the reviewers found the current description and validation of CaCLEAN inadequate. As described the technique is semi-empirical, choices of key parameters are not experimentally constrained, and there are not enough data to evaluate the technique's accuracy in identifying the true locations of Ca^2+^ release events. To address these issues, the following essential concerns must be adequately addressed before the paper can be accepted.

We thank the reviewers for their thorough and positive comments and the very constructive criticism. We have taken all point on board and have revised the manuscript accordingly. Please find our point-by-point response to the reviewers´ comments below.

1) CaCLEAN needs additional validation before it can be accepted as ground truth about calcium release events. The performance of the algorithm depends highly on underlying assumptions of parameters and the choice of diffusion coefficients. More confidence is needed that the location of release sites is an unbiased estimate of their true location, and that the presence or absence of a spark at a given location is not influenced by variations in dye loading, light intensity etc.a) The resolution limits of the CaCLEAN technique and implications for interpretation of results need to be discussed more quantitatively. What proportion of the CaCLEAN sites are individual as opposed to overlapping couplons? Thresholding is used in the CLEAN iterations to avoid false positives, but how does the choice of threshold affect the number of missed events (i.e., couplons)?One approach to give a more complete description of the accuracy and limits of the technique would be to simulate dye images of large numbers of closely localized simultaneous sparks at known locations, and determine how precisely and accurately those locations are recovered by the CaCLEAN process. Ideally this would involve a 3D reaction-diffusion model, but this may not be practical for a variety of reasons, including the time required, and the lack of consensus on a single calcium model with buffering for the myocyte.As a reasonable compromise, one could generate a single spark image using existing spark models, and linearly superimpose a large number of copies with various centers to see if CaCLEAN can recover the centers. This would address the major question of whether clustering and overlapping of the fluorescence patterns cause any systematic bias in CaCLEAN, even though it would not take into account all the nonlinear buffering interactions among sources. This should be do-able, preferably with the sparks placed in 3 dimensions and then "imaged" with the confocal PSF.In this way one could specify a range of critical conditions (couplon spacing, S/N ratio, assumed values for CLEAN thresholding and diffusion coefficients) required for the algorithm to generate an accurate map, and to place quantitative limits on its accuracy. Such information will be essential to validate the technique and to enable others to use it properly.

We agree with the reviewers that the resolution limits of CaCLEAN are really important and should be thoroughly evaluated. We address this issue in three different ways.

1) We simulated two local Ca^2+^ transients (Figure 1—figure supplement 5) with various distances between their centers. These events were convoluted with noise levels also observed in our original recordings and passed through the CaCLEAN algorithm. From these data we concluded that the CaCLEAN algorithm is able to resolve local transients whose centers are 1 µm apart (Figure 1—figure supplement 5). In rat ventricular myocytes the mean distance between RyR clusters was reported to be 1.02 µm (Soeller C, et al., PNAS 2007, 104:14958). The new figures are included and appropriately referenced in the main text.

2) Following the reviewers´ suggestions, we simulated a Ca^2+^ transient with gridded Ca^2+^ release (Figure 1—figure supplement 6Aa and b). After adding realistic recording noise (Figure 1—figure supplement 6Ac) these data were passed through the CaCLEAN algorithm and couplon maps were generated (Figure 1—figure supplement 6Ad). Superimposing the calculated maps with the original simulate maps (Figure 1—figure supplement 6B) demonstrated that a good correlation between these two data sets. The new figure has been added and appropriately referenced in the main text.

3) As suggested by the reviewers, we generated a single Ca^2+^ release image with a descriptive model, as described in the Materials and methods section (“Simulation of the upstroke phase of a Ca^2+^ transient with random Ca^2+^ release sites”), and distributed 1000 copies of such events randomly inside the circumvent of a typical ventricular myocyte (Figure 1—figure supplement 7). This new figure was added and appropriately referenced in the main text. The resulting data was passed through the CaCLEAN algorithm (Figure 1—figure supplement 7). Superimposing the centers of Ca^2+^ release centers with the CaCLEANed couplon map (Figure 1—figure supplement 7) demonstrated that CaCLEAN algorithm did neither generate any false positive ECC couplons and nor lose any Ca^2+^ release events. Based on these tests in both xy- and z-axis (Figure 1—figure supplement 4), we believe that the CaCLEAN algorithm can reproduce physiological 3D distribution of active ECC couplons throughout rat ventricular myocytes.

b) As a further consistency check, it would be useful to use the algorithm to reconstruct release sites in quiescent myocytes that exhibit spontaneous sparks. Raw frame data for that analysis should be readily available (perhaps from frames between stimulations). This test should allow a visual comparison between observed sparks and ECC release sites which would provide a consistency check as well as demonstrate the small subset of couplons that are engaged in spontaneous activity.

We completely agree with the reviewers. Unfortunately, our recording conditions and the physiological occurrence of spontaneous events do not allow this task to be fulfilled. We have to use robust laser power (30 ~ 70 mW on the input, distributed over 2500 simultaneous excitation spots) to gain sufficient signal-to-noise ratio. Such high excitation levels are not tolerated by the myocytes over extended period of times, we therefore designed an experimental regime (custom build soft- and hardware) in which the myocytes are only illuminated for brief periods of time (around 100 ms) as detailed in Figure 4 and the Materials and methods section of our manuscript. In this way, we do not record any data other than the upstroke phase of all Ca^2+^ transients. Furthermore, the physiological frequency of Ca^2+^ sparks is rather low (100 events/s per cell, around 1.5 events/s per imaging plane; Cheng H, et al., Science 1993, 262:740). To reconstruct a meaningful ECC couplon map from intact myocytes, it would need hours or continuous recording to gather sufficient Ca^2+^ sparks (e.g. at least 1 sparks/μm^2^ with an optimal density of around 3 sparks/μm^2^). Such extended recording times appear not accomplishable, at least to our knowledge. We have tried to illustrate this in Author response image 1 based on our own experiments.

To reconstruct a good representation of release sites in quiescent myocytes, it would need 8000 ~ 10000 Ca^2+^ sparks per cell, i.e. at least 3 sparks/μm^2^.

c) The algorithm assumes symmetrical diffusion, raising the question of how the reconstruction would be affected by diffusion barriers closed to couplons, e.g., mitochondria. Ideally this would be addressed by using appropriate simulated data, but if this is beyond the scope of the current study, a brief discussion of such potential effects should be included.

We agree with the reviewers and have included an additional paragraph discussing this specific issue in the seventh paragraph of the Discussion.

2) Better quantification in several places is needed.a) Regarding the proximity of Ca^2+^ release sites to the plasma membrane in Figure 1, the text refers only to a "very good correlation." A quantitative measure should be included.

A quantitative measure, structural similarity index (SSIM) (Wang et al., IEEE Transactions on Image Processing 2004, 13:600), has been included as a new figure panel (Figure 1Dd) and references in the main text. We calculated the SSIM for structures that appear different by nature, such as the plasma membrane and the ECC couplon map. In other cases when comparing two sets of data that by nature are same, we always use the correlation coefficient.

b) In Figure 2, the masks show a "very good match" while in Figure 2 the pattern of couplons is "rather different." Important information could be gleaned by measuring the probability of firing for each couplon over multiple stimuli, revealing the location of couplons that are highly reliable vs. those that are not. Are the most reliable couplons the first to fire? The number of cells analyzed should also be stated.

A quantitative analysis of the correlations has been included as new panels in Figure 2Af and 2Bf. A probability map indicating the probability of firing for each couplon has been generated and included as a new Figure 2—figure supplement 1. From this, the reliability of the couplons can be deducted. However, due to the limited speed of imaging devices when compared to the physiological process, it is for us not feasible to readily resolve the firing order of the couplons during a specific Ca^2+^ transient. A previous report has characterized the property of coupling sites in atrial myocytes utilizing line scanning at 10.000 lines/s (Mackenzie et al., J Physiol 2001, 530: 417).

c) In Figure 1, Ca^2+^ signals are measured at different distances from entry sites, but they are not strictly comparable because they are recorded in the vicinity of different entry sites. It would be more informative to compare responses at variable distances from the same entry site, and to average several examples if possible.

As suggested by the reviewers, we have generated an entirely new Figure 1 employing indeed adjacent pixels of the very same ECC couplon site (detailed in Figure 1Ea). The corresponding text has also been modified accordingly.

3) The algorithm, and in particular the ACO, needs to be more clearly described. How is the ACO generated, what is the equation for it, and how are the particular choices of ACO parameters justified? Is the ACO amplitude or size the more important variable? How does the choice of ACO parameters (amplitude, diffusion coefficient) affect the lateral spatial resolution?

We have totally re-organized the Materials and methods section and have added several new paragraphs following the suggestions of the reviewers including a description of ACO.

In addition, we include a new paragraph discussing the choice of ACO values (Discussion, fifth paragraph).

4) The application of CaCLEAN to understand the negative amplitude-frequency relationship is a potential strength of the paper. Early on the authors mention decreased SR Ca^2+^ content and a decreased number of participating couplons as possible causes, but their respective contributions are not fully resolved here. A decreased number of couplons certainly contributes (Figure 3Cb), but are the Ca^2+^ transient amplitudes of active couplons constant or do they decline with frequency? One way to address this would be to calculate the mean and s.d. of Ca^2+^ signal time course from the entire ensemble of couplons at low and high driving frequencies, and plot the average F vs. time to illustrate any changes. Multiplying this waveform by the number of active couplons may then demonstrate whether these changes are sufficient to account for the observed Ca-frequency relationship.

Following the reviewers´ suggestions, we further analyzed the behavior of individual couplons. The amplitude and size of active couplons do significantly decline with increasing frequencies, as shown in the new figure panel Figure 3Cc and d.

To estimate the contribution of couplon CLEAN signal strength and density to the observed negative Ca-frequency relationship we calculated the couplon signal mass in analogy to spark masses (signal strength * FWHM) and eventually multiplied by the couplon density (/1000µm). The resulting relationship reflected the expected negative frequency relationship as depicted in Author response image 2. We refrained from including this into Figure 3 because it did not add substantial new information apart from a similar trend (cmp. to Figure 3Ca). The exact relationship majorly depends on the specific interpretation of the “couplon amplitude” (Figure 3Cc).

**Author response image 2. respfig2:** 

In the Abstract, the negative amplitude-frequency relationship is attributed to decreased firing reliability at higher frequencies. However, it is not clear whether release reliability (as defined by the temporal and spatial CV) tells us anything more than the decline in the total number of active couplons. This should be clarified.

We agree with the reviewers that it would be more comprehensive to provide additional details on this rather important and interesting aspect of our study in the Abstract. According to *eLife* rules the Abstract is limited to 150 words. Its current length is already 150 words. We have slightly turned the phrasing down to accompany also additional mechanisms contributing to the negative Ca-frequency relationship. It now reads “[…]stimulation *including* a decreasing and increasing firing reliability, respectively.”